# A neural network-based model framework for cell-fate decisions and development
Mátyás Paczkó [1,2,6], Dániel Vörös[1,2,6], Péter Szabó[1], Gáspár Jékely [3], Eörs Szathmáry [1,4,5] & András Szilágyi[1]

Gene regulatory networks (GRNs) fulfill the essential function of maintaining the stability of cellular differentiation states by sustaining lineage-specific gene expression, while driving the progression of development. However, accounting for the relative stability of intermediate differentiation stages and their divergent trajectories remains a major challenge for models of developmental biology. Here, we develop an empirical data-based associative GRN model (AGRN) in which regulatory networks store multilineage stage-specific gene expression profiles as associative memory patterns. These networks are capable of responding to multiple instructive signals and, depending on signal timing and identity, can dynamically drive the differentiation of multipotent cells toward different cell state attractors. The AGRN dynamics can thus generate diverse lineage-committed cell populations in a robust yet flexible manner, providing an attractor-based explanation for signal-driven cell fate decisions during differentiation and offering a readily generalizable modelling tool that can be applied to a wide variety of cell specification systems.

Genetic regulatory systems, which dynamically control developmental/ cellular differentiation processes, operate through activating and inhibitory interactions among sequence-specific transcription factors (TFs) and their target DNA sequence elements, known as cis-regulatory modules (CRMs), that determine when and where transcription occurs[1–4]. Activating and inhibitory interactions are highly combinatorial and lead to the formation of complex gene regulatory networks (GRNs)[3,5] which can be decomposed into subcircuits or functional building blocks that reflect the basic logic behind an individual component of an intricate developmental process[5–7]. These functional building blocks must, on one hand, provide stability for certain differentiation stages and, on the other hand, be able to drive the dynamics of the system toward transitions to other states in response to internal or external triggers, while controlling the residence time in different developmental stages[8–11].

However, despite substantial efforts to elucidate the core transcription factor subnetworks associated with different cell types[12] and the existence of a large number of theoretical and experimental studies on lineage choice, the regulatory roles played by the functional building blocks of GRNs in cell fate decisions have not yet been systematically and adequately mapped[13]. Therefore, the fundamental questions of how cellular states and transitions

between them are defined, and how environmental cues and cell-intrinsic machinery and their interplay govern these processes, remain elusive[12,14]. Waddington's epigenetic landscape concept[15] and the analogous energy landscape view emerging from network biology have had a profound impact on the conceptualization of cell fate decisions in this context. According to these insightful metaphors, a landscape consists of a series of branching valleys that contain a set of attractors, which represent temporally stable cellular states that are defined by the constellation of the genes characteristically expressed in these particular states[12,15]. Every theoretically possible cell state can then be characterized by an energy value depending on the state-specific expression levels (or, according to the classical Boolean representation, on/off statuses) of the genes considered in the system. Hence, from an energy-based viewpoint, an attractor cell state corresponds to one of the local energy minima-, or to the global energy minimum of the landscape, where the gene expression statuses are aligned according to the regulatory forces and these forces are consequently dissipated, as a result of which the dynamics is temporarily, or ultimately relaxed (i.e., reaches a steady state)[16,17]. From a more practical point of view, the most commonly applied method to infer the shape of the landscape is based on the calculation of the negative logarithm of the steady-state probability distribution of

[1]Institute of Evolution, HUN-REN Centre for Ecological Research, Konkoly-Thege M. út 29-33, 1121 Budapest, Hungary. [2]Doctoral School of Biology, Institute of Biology, ELTE Eötvös Loránd University, Pázmány Péter sétány 1/C, 1117 Budapest, Hungary. [3]Living Systems Institute, University of Exeter, Stocker Road 4QD, EX4 Exeter, UK. [4]Center for the Conceptual Foundations of Science, Parmenides Foundation, Hindenburgstr. 15, 82343 Pöcking, Germany. [5]Department of Plant Systematics, Ecology and Theoretical Biology, Eötvös Loránd University, Pázmány Péter sétány 1/C, 1117 Budapest, Hungary. [6]These authors contributed equally: Mátyás Paczkó, Dániel Vörös. ✉e-mail: szathmary.eors@ecolres.hu

the gene expression state space[18]. With this approach, the elevation of the landscape is determined by the inverse of the probability density function, as a consequence of which the states with the highest probability density will be characterized by the lowest potential[18,19]. This landscape paradigm has contributed to the development of a variety of dynamical models, examining cell differentiation and reprogramming processes from an attractor-based perspective[4,20–23]. For example, it has facilitated the construction of multi-dimensional energy landscapes of master regulator genes, based on Boolean logic operators that combine multiple input signals, thereby revealing key attractors and identifying potential reprogramming barriers[17]. However, the landscape view has its own limitations in terms of its potential to capture experimentally validated differentiation trajectories, as it has, so far and to some extent, failed to account for the relatively stable but still transitory intermediate cell types observed during differentiation[12,24]. This challenge is well illustrated by the fact that the majority of dynamical models of cell differentiation exhibit a mutually exclusive dichotomy between a dynamically stabilized state and inherent forward momentum[25,26].

Contrary to the bottom-up approaches of, e.g., chemical reaction networks[27] or Boolean network models, associative neural networks provide a top-down alternative to investigate the topological and dynamical properties of the functional building blocks of gene regulatory networks[28,29]. The key concept of this top-down approach is that associative memory within the context of developmental gene regulation – analogous to the conceptual idea of epigenetic landscapes – can be described by an energy descent dynamics[30] during which each gene expression (memory) pattern that corresponds to a certain cellular differentiation stage has a particular basin of attraction[31]. More specifically, the attractor feature of an autoassociative network means that it can solve the problem of recovering a particular state (usually represented as a vector), when presented with an initial pattern that resembles one of the memory vectors stored in its weights[30,31]. Thus, in response to an input pattern, such a network produces the same output pattern as the input, even if the input is burdened with some noise compared to the original pattern with which the network was trained. In hetero-associative networks, however, the input-output vector-pairs are different by definition[31]. Since the dynamical stability of, and change in, stage-specific gene expression during differentiation can be treated as auto- and hetero-associative memory pattern retrieval, the principles of associative neural networks can be applied to gene regulatory systems. Moreover, given a set of desired stable states (autoassociativity) or stage-pair transitions (hetero-associativity), the regulatory network of a given differentiation topology can be analytically determined by simple algebraic operations in the form of a regulatory weight matrix[32–34]. With this approach, the gene expression values across the differentiation stages will be ultimately determined by the regulatory matrix and a shared activation function that nonlinearly maps the summed regulatory effects (weights) of all genes into expression values.

However, extant models of developmental gene regulation utilizing the associative properties of neural networks have investigated this phenomenon only in the context of single stage-pair transitions[28,29], or development of environment-specific adult stages from a particular embryonic stage[31], without taking into consideration intermediate developmental stages and their stage-specific gene expression patterns. We extend the associative network-based description of GRNs to complex developmental processes by proposing an associative GRN model (AGRN) in which the functional key components of the regulatory mechanism are based on the appropriate combinations of elementary associative rules. We show that this model can accurately reproduce empirically observed developmental trajectories including intermediate stages with their corresponding stage-specific gene expression profiles. In terms of Waddington's epigenetic landscape view[12,15], we demonstrate that the modeled developmental stages can be characterized by attractor properties which enable the developmental or differentiation processes to reside in a certain basin of the landscape for a specific time period. We also present the simple mathematical framework which allows us to phenomenologically describe the transition mechanisms by which external signals can exert a lifting effect on the system residing in a basin and provide forward momentum to the developmental trajectory to progress toward other attractors.

Below we summarize the key concepts of our modeling techniques and introduce the terminology that will be used in the following. We consider three biologically important stage transitions: autonomous transition between two stages (linear transition), divergence into different stages (fork transition), and trigger-induced linear transition (conditional transition). Each fork transition has a default branch, which is the branch that the system will follow in the absence of a trigger, and a triggered branch, which the system will follow when the trigger is enabled. The existence of such a default output has been suggested, for example, in the case of hematopoietic stem cells (HSCs), which, in the absence of instructive signals, are thought to differentiate into macrophages, an evolutionarily ancient, default blood lineage[14]. Triggers model external cues (e.g., mechanical) or signaling factors and could be singular or repetitive as e.g., during binary fate specification[35,36]. The combination of these transitions allows us to describe almost any kind of biologically plausible topologies in cell-differentiation trajectories. We model gene expression changes following the associative dynamics of Vohradsky and Szilágyi et al.[28,29,31], where the state of the system at time $t$ can be described by a gene expression vector $\mathbf{p}(t) = (p_1, p_2, ..., p_N)^T$, with its elements representing the expression levels of different genes. To model the time evolution of the gene expression, we construct a regulatory matrix $\mathbf{M}$, in which entry $m_{ij}$ defines the regulatory effects: positive/negative values indicate that regulatory unit $j$ has a direct or indirect activating/inhibitory effect on another regulatory unit $i$ (see refs. [37,38]), so that regulators can also be regulated and units represent genes and/or epigenetic elements. The regulatory matrix for an elementary stage transition (linear, fork or conditional) can be constructed by the developmental stage vectors of the initial and final stages of the given transition and the triggers. A developmental stage vector is extracted from empirical data and it represents the gene expression profile of a given stage (Fig. 1a). Note that while developmental stage vectors are (constant) binary valued (on/off) vectors, the time-dependent gene expression vector is continuous valued. The regulatory matrix of a complete differentiation hierarchy results from the summation of the matrices that implement the elementary stage transitions included in the hierarchy within each of its alternative pathways (Fig. 1b, c, Supplementary Note 1). Such a matrix can then dynamically regulate the expression states of the individual genes throughout the differentiation process in an autonomous fashion and is therefore referred to as a regulatory program matrix. Thus, a regulatory program matrix, the developmental stage vectors and the differentiation topology from which the matrix is constructed serve as the model input (Fig. 1a, b), while the time series of gene expression levels, characterizing the differentiation stages and determined by the regulatory program matrix and the triggers, are the model output (Fig. 1c, d). For a detailed mathematical description of the model, see Methods.

## Results
### AGRN model of hematopoiesis
To understand how auto- and heteroassociative rules implemented into the AGRN model based on empirical data can reproduce experimentally validated dynamical gene expression, we first use our framework to analyze a human hematopoiesis dataset (Supplementary Data 1). In this dataset, we defined stage-specific gene expression profile vectors (developmental stage vectors) for the cellular stages of the hematopoietic hierarchy (see Methods for details). The differentiation topology we consider here[39] consists of 13 differentiation stages (i.e., cell states), which are modeled by a combination of signal-driven binary cell fate decisions (represented by fork transitions) and autonomous linear transitions, and one conditional (signal-driven linear) transition (Fig. 2a). Note that as hematopoietic differentiation and cell division events are shown to be temporally separated[40,41], the arrows between the differentiation stages at fork transitions represent the potential transition directions, rather than asymmetric cell divisions. An extracellular signaling mechanism, which regulates the maintenance of the quiescent state of long-term repopulating hematopoietic stem cells (LTR-HSCs) and their transition to the active short-term repopulating hematopoietic stem

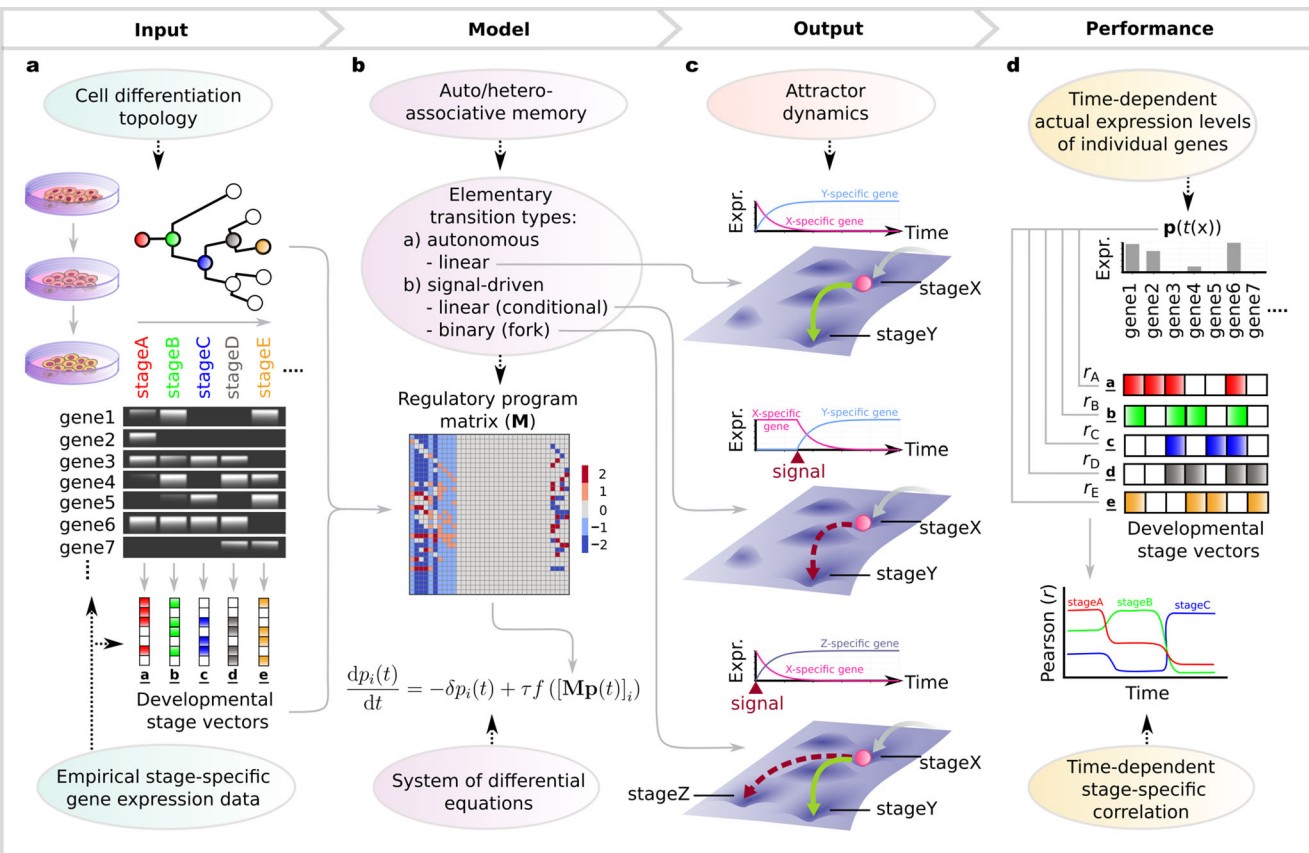

**Fig. 1 | Schematic illustration of the AGRN model. a** Input components of the AGRN framework. An AGRN model takes as model inputs: (i) the cell differentiation topology of a given developmental process, i.e., the order of the subsequent developmental stages corresponding to different cellular identities, and (ii) the stage-specific binary representations of individual gene expression states in the form of developmental stage vectors. In the developmental stage vectors, colored cells represent on and empty cells represent off gene expression states. The input components are then used to construct a regulatory program matrix **M** according to simple, modular algebraic rules described in Eqs. (4–6). The **M** matrix then governs the dynamics of the model by regulating the expression levels of individual genes ($p_i$) as described by the system of differential equations in Eq. 1. (**b**). **c** Output components of the AGRN framework. The attractor dynamics is realized by a series of elementary transitions encoded by the associative memory of the matrix. At fork and conditional transitions, in concert with cell-intrinsic machinery (represented by the regulatory program matrix), instructive external signals (triggers) determine the behavior of the system. Pink marbles represent the system's current state, continuous green- and dashed red arrows represent default- and trigger-induced differentiation pathways, respectively. Gray arrows with blurred end represent previous transitions. The elementary transitions correspond to time series of gene expression level changes (here, for simplicity, only the expression of one key gene per stage is shown). To measure the performance of the model over time, we calculated the Pearson correlation coefficients ($r$) between the state of the gene expression vector ($\mathbf{p}(t)$) and each developmental stage vector at all $t + \Delta t$ time points (**d**).

cell (STR-HSC) stage[42,43], is incorporated into the model by the conditional transition between these two stages, where expression of an external signal-mediating trigger ($tr$-$3+$) induces the transition. Thus, this transition type provides a means to dynamically control the residence time in the quiescent LTR-HSC stage. Firstly, the model performance on the data is measured by a set of Pearson correlation coefficients between the $\mathbf{p}(t)$ expression vector (vector for the actual dynamical expression state of the genes) and the developmental stage vectors. Figure 2b shows this measure as a function of the time in case of two illustrated differentiation pathways. The left panel shows that the differentiation process follows the Mesoderm→CLP pathway as initial and terminal differentiation stages, if three external signals—mediated by the respective triggers ($tr$-$1+$, $tr$-$2+$ and $tr$-$3+$)—are presented at the appropriate time (denoted by vertical arrows at the x-axis). The right panel shows that the differentiation process follows the Mesoderm→CFU-E pathway, if three additional external signals are mediated ($tr$-$4+$, $tr$-$5+$ and $tr$-$6+$) at the right time. Consistent with this, principal component analysis of the model shows that linear combinations of consecutive samples from the $\mathbf{p}(t)$ expression vector converge to the stages of the hematopoietic hierarchy (i.e., to the pre-defined developmental stage vectors) in an appropriate order (Fig. 2c). We found that the gene expression dynamics of this system driven by a modular AGRN regulatory network

(i.e., the same differentiation topology and developmental stage vectors, with the only difference being that the dynamics is governed by three different regulatory program matrices) results in a qualitatively similar performance (Supplementary Note 4, Fig. S3).

## AGRN model of cell cycle

The combination of the elementary associative rules of the proposed framework enables us to describe cyclic dynamics as well. This property is a critical requirement for developmental gene regulation models, considering that the cell cycle is a major determinant of the temporal gene expression patterns on a cellular level[44,45]. To demonstrate this model property, we assembled a human cell cycle (CC) dataset (Supplementary Data 2) which consists of phase-specific gene expression profiles for the four CC phases and the associated apoptotic process (see Methods). Using this dataset, we demonstrate that expression timing of individual genes, which are involved in the CC dynamics and thus constitute the $\mathbf{p}(t)$ expression vector, exactly follow the genes' corresponding CC phases in a cyclic fashion (Fig. 3a, b). We also show that, by implementing fork transitions, the model can accurately describe a termination of the cyclic dynamics promoted by an external signaling mechanism (Fig. 3c), where an apoptotic signal is mediated by the expression of a trigger ($tr+$) that causes the system-level

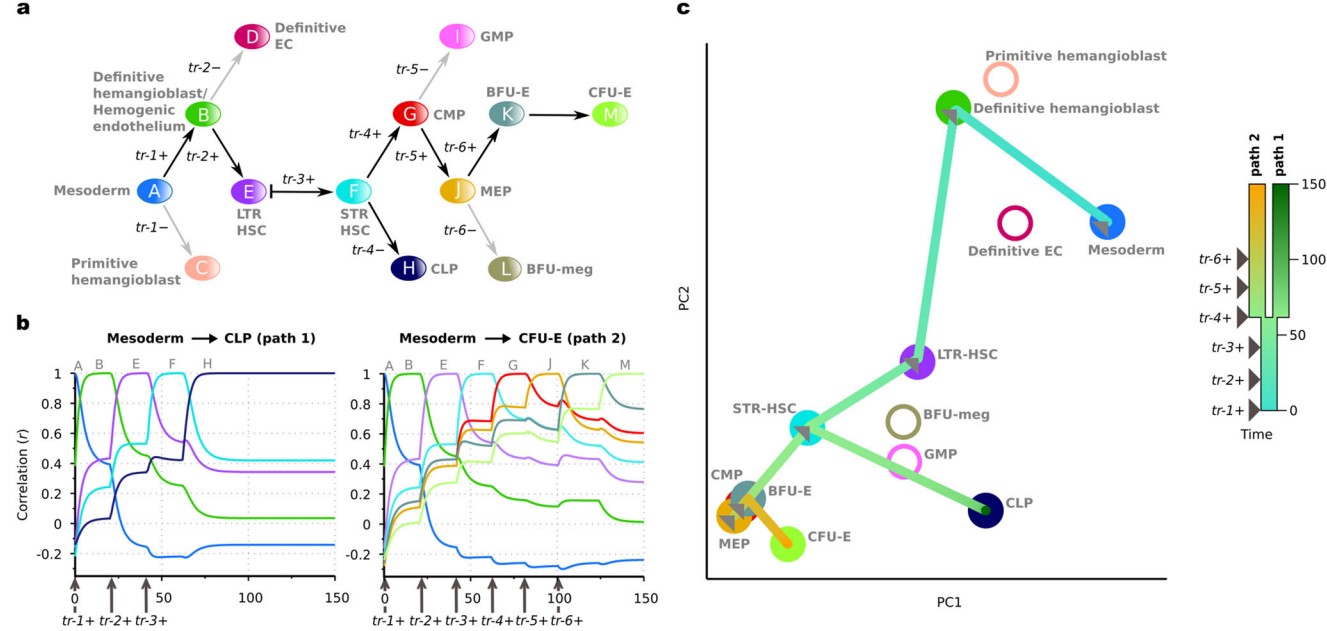

**Fig. 2 | Illustration of the hematopoietic cell differentiation process with an associative GRN. a** Hematopoietic differentiation topology of the model. Uppercase letters with rounded and colored background represent cell differentiation stages, the arrows between them represent transitions between the stages. The fork transitions and the conditional transition are controlled by the expression of transition-specific triggers (denoted as *tr-1, tr-2, …*). The two differentiation pathways demonstrated here are highlighted with black arrows. **b** Realizations of stages, as measured by Pearson correlation coefficients between the **p**(*t*) expression vector and the stage-specific developmental stage vectors (Supplementary Data 1). Color code for the lines that correspond to the cellular differentiation stages is given at (**a**). **c** The first two principal components of the differentiation stages of the hematopoietic

hierarchy and the dynamical trajectories of the system. Principal components for the stages are obtained from the developmental stage vectors. PCA trajectories of the two realized pathways are obtained from the **p**(*t*) expression vector sampled at $\Delta t = 0.1$ frequency. Color code for the time scale of consecutive samples with the timing of the corresponding triggers is shown on the right. In the nomenclature and topology of the differentiation hierarchy, we followed ref. 39. Abbreviations: EC endothelial cell, LTR long-term repopulating, HSC hematopoietic stem cell, STR short-term repopulating, CMP common myeloid progenitor, CLP common lymphoid progenitor, GMP granulocyte-macrophage progenitor, MEP megakaryocyte erythroid progenitor, BFU-E burst forming unit erythroid, BFU-meg burst forming unit megakaryocyte, CFU-E colony-forming unit erythroid.

gene expression pattern (Fig. 3d, e) to irreversibly diverge from the phases of the cycle and to converge toward an alternative (terminal) fate.

### AGRN model of *Caenorhabditis elegans* embryonic development

Due to its well-known developmental pathways and stage-specific gene expression patterns, the *Caenorhabditis elegans* embryonic development is an ideal process to test the AGRN model functionality on a larger differentiation topology with a considerably higher number of stages and genes. For this purpose, we assembled a *C. elegans* embryonic development dataset (Supplementary Data 3) that consists of 2435 genes corresponding to 1046 cellular differentiation stages (Fig. 4a, Methods and ref. 46). With large datasets like this, where a considerable amount of different, often conflicting associative rules are implemented into the regulatory program matrix which could pose a serious difficulty on the regulatory functionality of the system, our aim is also to see to what extent the performance of the AGRN model changes relative to that of the more simple systems analyzed above (i.e., the human hematopoiesis and cell cycle models). Notably, the model successfully describes the gene expression changes of the illustrated differentiation pathways with one regulatory program matrix and without a substantial deterioration in the performance relative to more simple systems (Fig. 4b). Principal component analysis of the model shows that linear combinations of consecutive samples from the **p**(*t*) expression vector converge to the *C. elegans* embryonic developmental stages (i.e., to the pre-defined developmental stage vectors) in an appropriate order (Fig. 4c).

### Alternative trajectories

Cell-lineage differentiation is often perceived as a hard-wired process but, contrary to this notion, reprogramming studies suggest that differentiating cells can be remarkably plastic in terms of their cellular identity changes[12,47].

Even in terminally differentiated cells, it is possible to wake up dormant gene expression programs, meaning that with the right set of transcriptional factors or signals (triggers), developmental stages can be switched into each other[10,48]. In our model, if the signal corresponding to the triggered branch is activated after the transition to the default branch, the developmental pathway may converge to the triggered branch, thereby going through an alternative pathway, where the initial stage of a certain transition is followed by the default then the triggered stage. This alternative transition can be utilized even numerous forks later, or between distant forks as well, if the two expression states are not too much different. This means that in case of natural GRNs, the chance of a successful alternative transition decreases with topological distance as expression profiles of the stage vectors diverge during development.

In order to demonstrate that these alternative developmental pathways are possible to achieve by the AGRN model, we recreated the alternative routes shown in ref. 49. (see Fig. 5). We concluded that most of the alternative pathways from the reference model are accessible in our model framework. Three pathways are unattainable, as they are default cell fates. This shows that this model provides the correct amount of flexibility to describe natural cell differentiation processes.

### Robustness against perturbations

In order to dissect the behavior of the AGRN model framework in case of perturbations, we analyzed the consequences of the following two different types of perturbations on the model performance: (i) multiplicative and nullifying perturbation of regulation strengths in the regulatory program matrices, which can be interpreted as perturbed interactions among transcriptional factors (e.g., by mutation of binding sites); and (ii) perturbation of the expression vector with mistimed gene expression, which can be

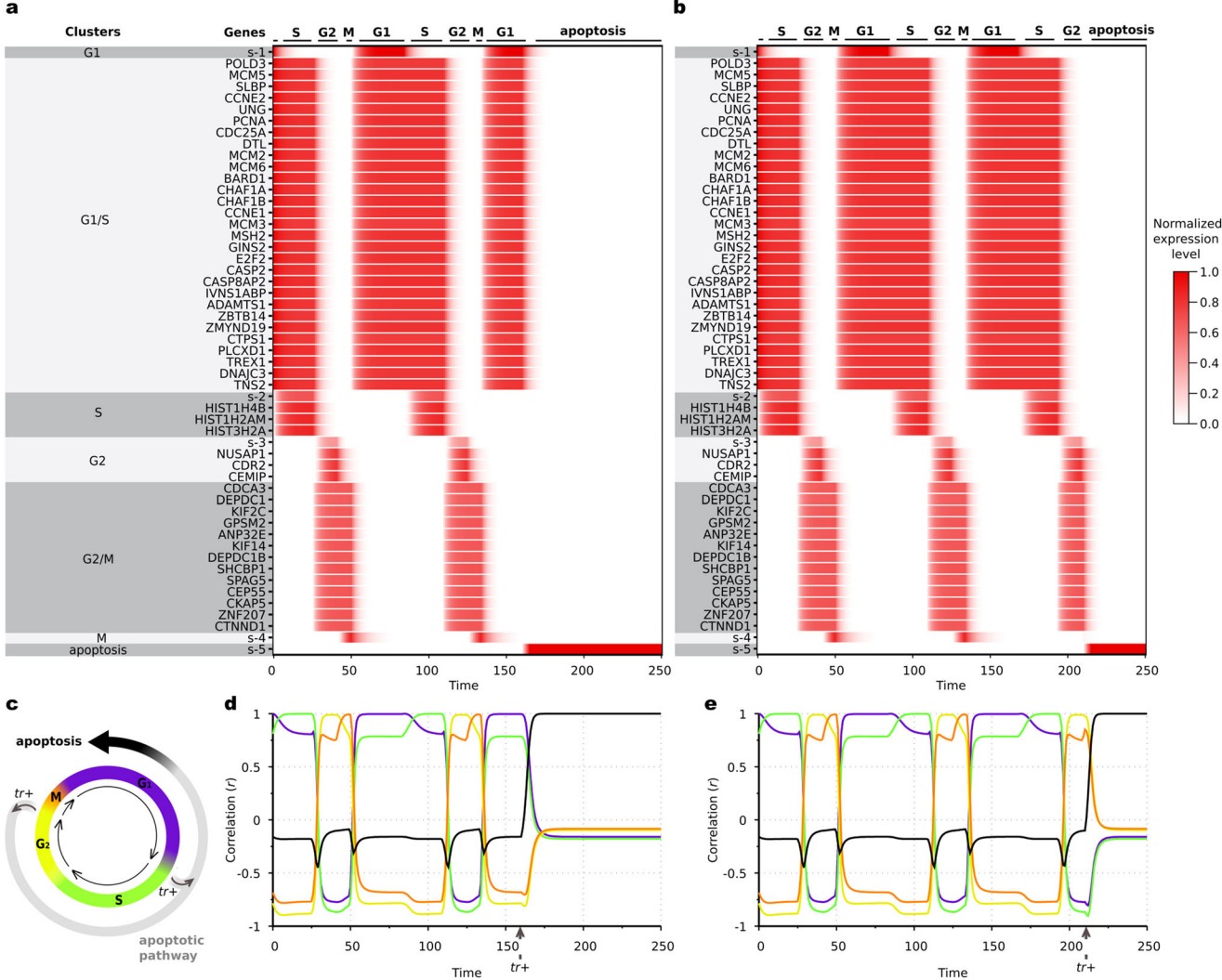

**Fig. 3 | Demonstration of the AGRN model functionality to describe cyclic dynamics on the human cell cycle (CC) data. a, b** Gene expression time evolution of individual genes in the **p**(t) expression vector. Expression levels are normalized by the maximal gene expression rate ($\tau/\delta$) of the model. **c** Positions of the checkpoints considered in the cycle. **d, e** Realization of phases, as measured by Pearson correlation coefficients between the **p**(t) expression vector and the cell cycle phase-specific gene expression profile vectors (Supplementary Data 2) as a function of the time.

Note the common x-axes with (**a**) and (**b**). Color code for the lines that correspond to the phases is given at (**c**). Termination of the cyclic dynamics and converging into the subsequent apoptotic pathway after two complete cycles is triggered by the expression of an apoptotic signal-mediating trigger (tr+) at the G1/S (**a, d**) and at the G2/M (**b, e**) checkpoints. The exact expression times of the triggers are denoted by vertical arrows on the horizontal axes.

interpreted as injecting a complete set of gene products from an other cell residing in a different stage, i.e. partial cytoplasm fusion. As Fig. 6a shows, the *C. elegans* P5.p vulval precursor cell differentiation (Fig. S4) and the human cell cycle (Fig. 3) models are exceptionally robust against multiplicative perturbations. In contrast, the hematopoietic model system, which incorporates a larger number of fork transitions (Fig. 2a), has a steeper drop in the performance at small perturbation strengths. In general, the performance of the model systems decreases slowly with increasing perturbation; even if $\sigma = 5$, 80% of the simulations go through the proper pathway without error. The biologically more implausible nullifying perturbation type (Fig. 6b) is more adverse; zeroing 2% of elements of the regulatory program matrix halves the performance. Note that the differentiation topologies of different size and complexity are of remarkably similar characteristics of performance.

We also analyzed the effects of misexpression of cellular identity determining key genes (i.e., mistimed expression of developmental stage vectors in the **p**(t) expression vector) on the dynamics. We found that following such perturbations, the characteristic behavior of the

hematopoietic system (Fig. 7a) is a typical down-regulation of the subsequent stages. However, at the same time, the system demonstrates substantial robustness against perturbations with regard to the convergence and stability of the target stage and other stages, non-proximal to the perturbation sites, see Fig. 7b, c. This behavior is independent of the timing of the misexpression (in the beginning, in the middle, or at the end of the expression of a given stage).

## Discussion

We have shown how the neural network-inspired associative approach to GRNs[28,29,31] allows the construction of arbitrarily large networks with required properties regarding the trajectories and rest points of important developmental processes. From a technical point of view, the question arises whether the black-box treatment of the common regulation function $f$ is sufficient or not, given that it can be expressed by different formulae with different parameters for different genes[50–52]. Here we used a scaled sigmoid-type activation function widely used in theoretical neuroscience (the original context of this dynamics), but its applicability to genetic regulatory systems

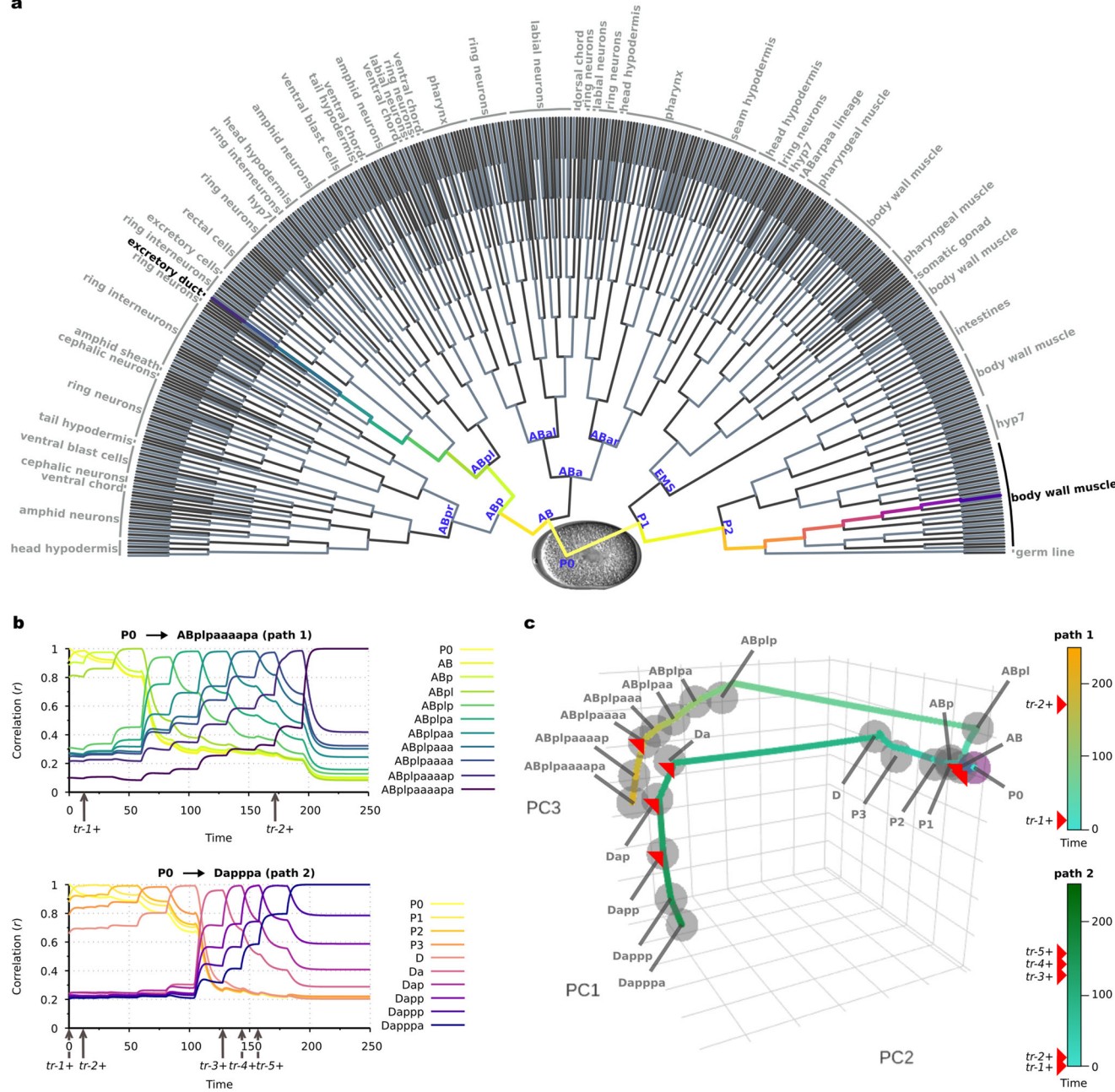

**Fig. 4 | *C. elegans* embryonic development with an associative GRN.**
**a** Differentiation topology of the model representing developmental stages and pathways. Black lines indicate linear transitions or the default branches of fork transitions, while gray lines illustrate the triggered branch of forks. The text labels at the tip of the tree indicate the tissue type that develops from the lineages. In the nomenclature and topology of the differentiation hierarchy, we followed ref. 46. **b** Realizations of stages, as measured by Pearson correlation coefficients between the $\mathbf{p}(t)$ expression vector and the stage-specific developmental stage vectors (Supplementary Data 3). **c** The first three principal components of the differentiation stages and the dynamical trajectories of the system. Principal components for the stages are obtained from the developmental stage vectors. PCA trajectories of the two realized pathways are obtained from the $\mathbf{p}(t)$ expression vector sampled at $\Delta t = 0.1$ frequency. Color codes for the time scale of consecutive samples with the timing of the corresponding triggers are shown on the right.

is likely an oversimplification as there is a plethora of different genetic regulatory interaction types.

Consistent with Waddington's epigenetic landscape view, the dynamical approach to development adapted by the AGRN framework proposes a generative model of gene expression changes upon differentiation based on attractor properties of certain stages[12,15]. Given an energetic or epigenetic landscape, a long-standing question is whether the landscape is static or not; in other words, whether cell fate decisions at critical points of a differentiation process are driven by noise or signals[13,18]. Although purely noise-driven cell fate decision modes have been the subject of serious debates and

the eligibility of the sharp dichotomy between signal and noise-driven modes has been questioned[53,54], a few studies pointed out that some cells may exist in an essentially stationary landscape and the main driving force of their differentiation is gene expression noise[13,55,56]. In contrast, it has been suggested that the landscape itself is dynamic; recurrently distorted by extrinsic signals that tightly regulate lineage commitment through several potential feedback mechanisms, thereby providing homeostatic control with a flexible means to quickly adjust the cellular output according to the needs of the organism[13,57–59]. While recognizing the potentially important role of gene expression noise in cell-fate decisions, the present study focused

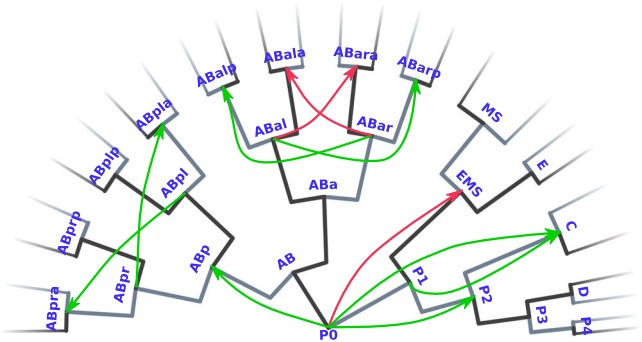

**Fig. 5 | Possible alternative pathways in *C. elegans* embryonic development.** The chart represents the differentiation topology of *C. elegans* embryonic development with the corresponding developmental stages and pathways. Black lines indicate the default branches of forks transitions, while gray lines illustrate the triggered branch of forks. Colored arrows indicate alternative pathways illustrated in ref. 49, which can be interpreted in our model. Color of the arrows represent reproducibility: greens are feasible, while red arrows illustrate the alternative pathways that are not achievable.

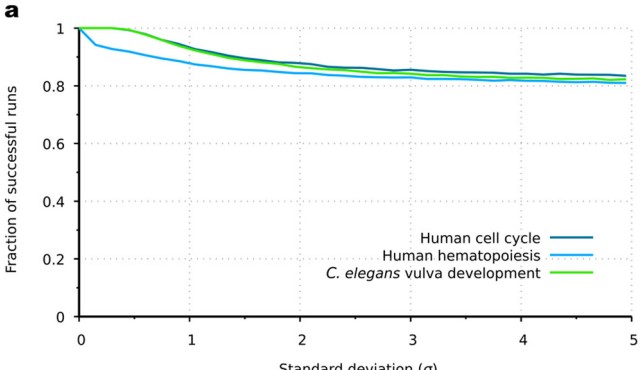

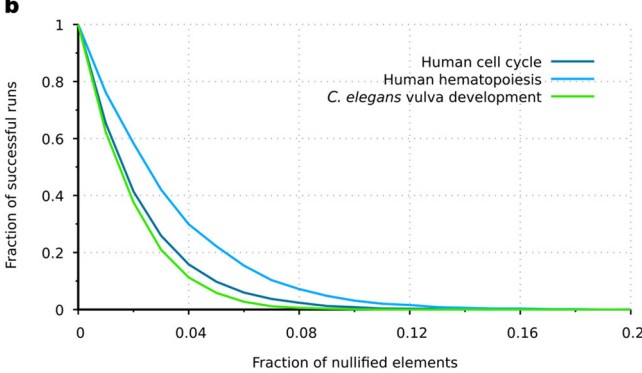

**Fig. 6 | The effects of regulatory interaction perturbations.** The performance of three AGRN model systems (human hematopoiesis, human cell cycle and *C. elegans* P5.p vulval precursor cell differentiation) as a function of the standard deviation ($\sigma$) of multiplicative perturbations (**a**); and as a function of the proportion of nullified elements (**b**) in the regulatory program matrices. The performance was measured as the fraction of successful runs (for detailed explanation, see Methods). Parameters are from the standard parameter set.

on the behavior of signal-driven, deterministic and tightly regulated systems. Regulatory program matrices in our model, constructed from empirical stage-specific gene expression vectors, are capable of completely reproducing each possible alternative differentiation program within a given differentiation topology in an autonomous manner, while incorporating a certain level of sensitivity for external cues (trigger-induced transition directions), thereby providing plasticity[60] for a particular developmental process. Our results therefore fit into a broader picture of cellular

differentiation as a process in which the interplay between environmental cues and cell-intrinsic machinery acts in a manner that (i) multipotent cells simultaneously exhibit co-accessibility of multiple lineage programs and have in place transcriptional circuits capable of responding to multiple extrinsic signals[14,61], (ii) gene expression noise is not a necessary condition for the corresponding gene regulatory networks to be able to generate diverse lineage-committed cell populations (i.e., drive the dynamics to different attractors) in a robust and yet flexible manner, thereby underpinning the role of dynamic, signal-driven landscapes in cell fate decisions[13]. A more thorough future investigation on the structure of the regulatory interactions among the elements in the AGRN regulatory program matrices – which describe not only direct gene-gene regulatory interactions, but rather they represent composite regulatory effects of genes, TFs, proteins[62], and epigenetic elements – may give a further insight into what kind of network features, such as the frequency of different motif (subcircuit) categories (see Supplementary Note 6 and Supplementary Table 1 in the present study) could be associated with the attractor properties of the dynamics and to what extent these network features as structural design principles are dependent upon certain cell differentiation topologies. Such investigations may help to better understand the regulatory principles behind these developmental processes, for example, by providing a means to categorize the corresponding regulatory networks into different network classes[6].

One possible future application of the AGRN approach relates to reprogramming studies (for a review, see: ref. 63), aiming to find potential transdifferentiation pathways and predict their feasibility by utilizing the attractor properties of cell differentiation landscapes[17]. In this context, our investigation on the attractor pool sizes in the hematopoietic cell differentiation hierarchy suggests that the definitive endothelial cell stage can be characterized by the largest basin of the landscape, as this is the stage into which the system-level gene expression pattern (the $\mathbf{p}(t)$ expression vector) converges most frequently in response to different mistimed triggers and perturbed genes (Fig. S1). We emphasize, however, that the latter statement is valid only under the assumption of the presence of these disruptive factors, which result in alternative differentiation trajectories, representing available reprogramming pathways (Supplementary Note 2).

From a broader perspective, our framework is relevant for simulating embryo-scale developmental processes and more generally toward developing a theory of development. An essential component of such a theory is a model to simulate gene-expression trajectories across cell lineages. Our framework achieves this for arbitrarily large differentiation topologies and their corresponding binarized gene-expression profiles, with a natural applicability to genetic and epigenetic regulation of gene expression[64,65]. We also suggest that our model can be used as plugin into more detailed spatial cellular models integrating GNRs with morphogenesis. Our model can also be refined by fitting the activation function to experimental data for specific gene families or individual genes. The AGRN approach can also be useful to synthetic biologists aiming to construct complex, but still robust network topologies[66], or to find biologically-inspired artificial circuits with special dynamical properties[67]. The AGRNs seem to have a useful balance between simplicity and complexity in that they offer a scalable tool to account for complex behavior.

## Methods
### Gene expression dynamics
Following Vohradsky and Szilágyi et al.[28,29,31] we formulate gene expression dynamics using associative networks formalism. Consider an organism with $N$ genes, and let us represent the expression state of the system (on a cellular or individual level) at a particular time $t$ by a vector $\mathbf{p}(t) = (p_1, p_2, ..., p_N)^{\mathrm{T}}$ with each element being the quantity of the product of a gene. The dynamics can be described by the differential equation (see refs. 28, 29,31)

$$\frac{\mathrm{d}p_i(t)}{\mathrm{d}t} = -\delta p_i(t) + \tau f\left(\left[\mathbf{M}\mathbf{p}(t)\right]_i\right) \tag{1}$$

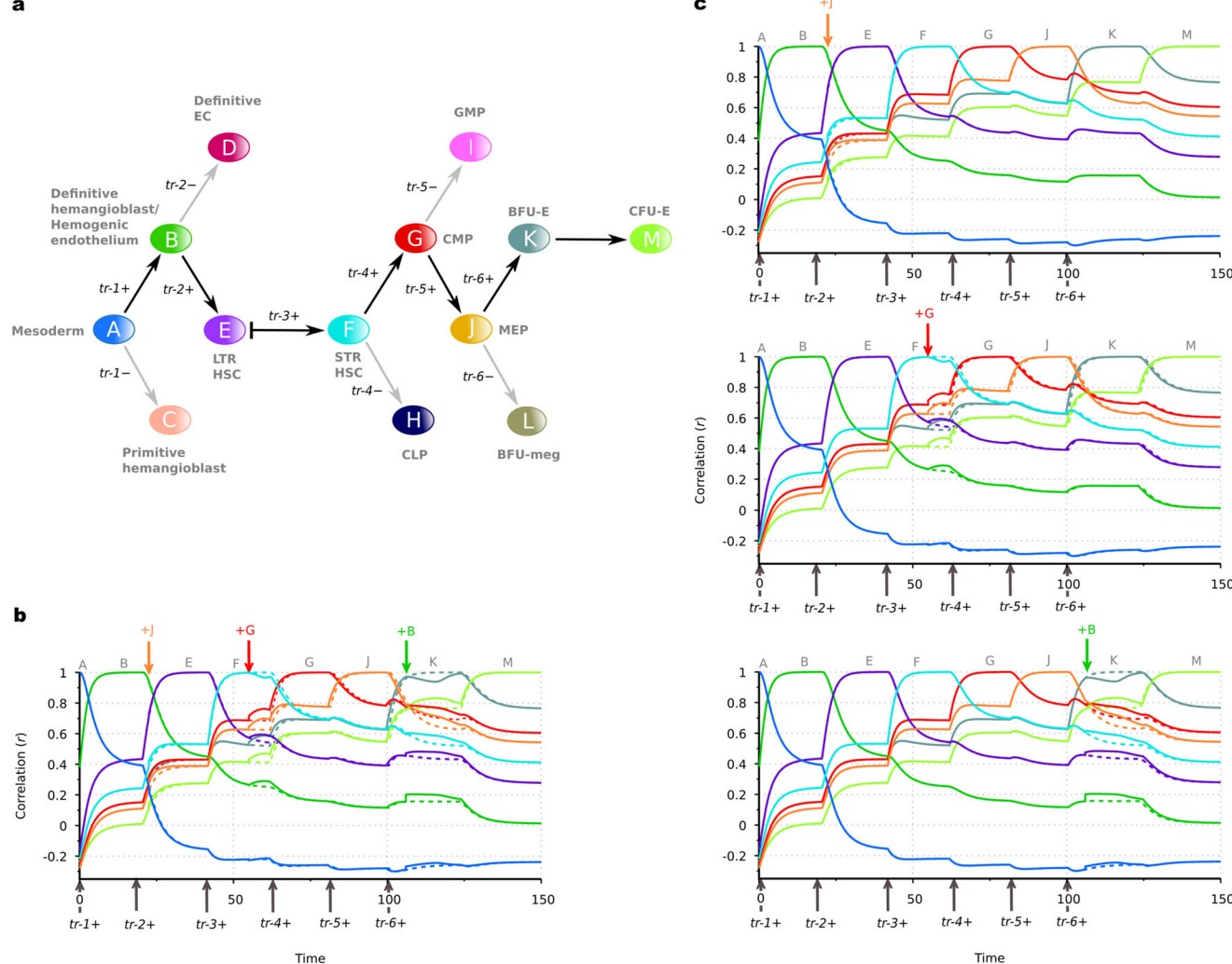

**Fig. 7 | Subsequent stage transitions with expression level perturbations. a** The structure of the differentiation topology. **b** Perturbation with three consecutive misexpressions: the first at the end of an expressed stage ($t = 22$), the second in the middle ($t = 55$), and the third at the beginning of an expressed stage ($t = 108$). **c** Single perturbations of the system with the same misexpression and at the same time as in (**b**). The notation and the color of the misexpressed stages are in line with (**a**). The level of misexpression is 5% of the maximal expression level ($0.05 \cdot \tau/\delta$). Dashed lines show the correlations of the unperturbed system (same as in Fig. 2b right panel), continuous lines indicate the perturbed ones. For the nomenclature and topology of the differentiation hierarchy, see Fig. 2.

where $\delta$ denotes the decay rate of gene products, $\tau$ denotes the maximal gene expression rate, $f(.)$ is the activation function, and regulatory program matrix **M** represents the gene regulatory network (see refs. 37,38)

The $m_{ij}$ elements of this regulatory program matrix define the pairwise regulatory effects between regulatory units: positive/negative values indicate that regulatory unit $j$ has a direct or indirect activating/inhibitory effect on regulatory unit $i$. The overall regulatory effect on any single gene is determined by the scalar product of the gene expression vector and the corresponding row of the regulatory program matrix, which is then mapped through a nonlinear activation function, in our model $f(x) = [1 + \tanh(\omega(x + \xi))]/2$. Here $\omega$ and $\xi$ are the scale and shift parameters of the activation function, respectively. According to Eq. 1, in equilibrium, each element of the expression vector can be either $p_i = 0$ (no expression) or $p_i = \tau/\delta$ (maximal expression). If not stated otherwise, we used the following standard parameter set: $\tau = 1$, $\delta = 0.2$, $\omega = 50$, $\xi = 0.05$.

For the representation of the expression profile of certain developmental stages that the system goes through we use {0,1}-membered (binary) stage vectors (hereafter developmental stage vectors), where 1 denotes that the given genes are expressed in the relevant stage. The organization of these vectors are the following: the head part contains stage-specific genes (a single gene for each stage that is expressed only in the given stage); the next vector part includes all the other genes that can be expressed in one or more stages and the tail of the vector contains triggers that govern the system (see later). Stage-specific genes are necessary for reliable operation, especially when the expression profiles of some developmental stages are similar (if there is no such a unique gene for a given stage in the empirical data, one has to introduce an artificial one).

Note that this organization of the developmental stage vectors is just for clarity and does not alter the outcome of the simulations. In the following the stage vectors will be denoted by **x**, **y**, **z**, . For the simple formalization of the model $\tilde{\mathbf{x}}$, $\tilde{\mathbf{y}}$, $\tilde{\mathbf{z}}$, . . . denote the modified version of the stage vectors, where the elements of the middle part are set to zero (the stage-specific and trigger elements will be unchanged). The regulatory program matrix **M** is formalized with the help of these vectors.

**Associative rules of the AGRN model**

The associative feature of the system means that given two stages X and Y, with the corresponding binary gene expression vectors **x** and **y**, it is possible to derive a regulatory matrix that initiates a transition from X to Y. The

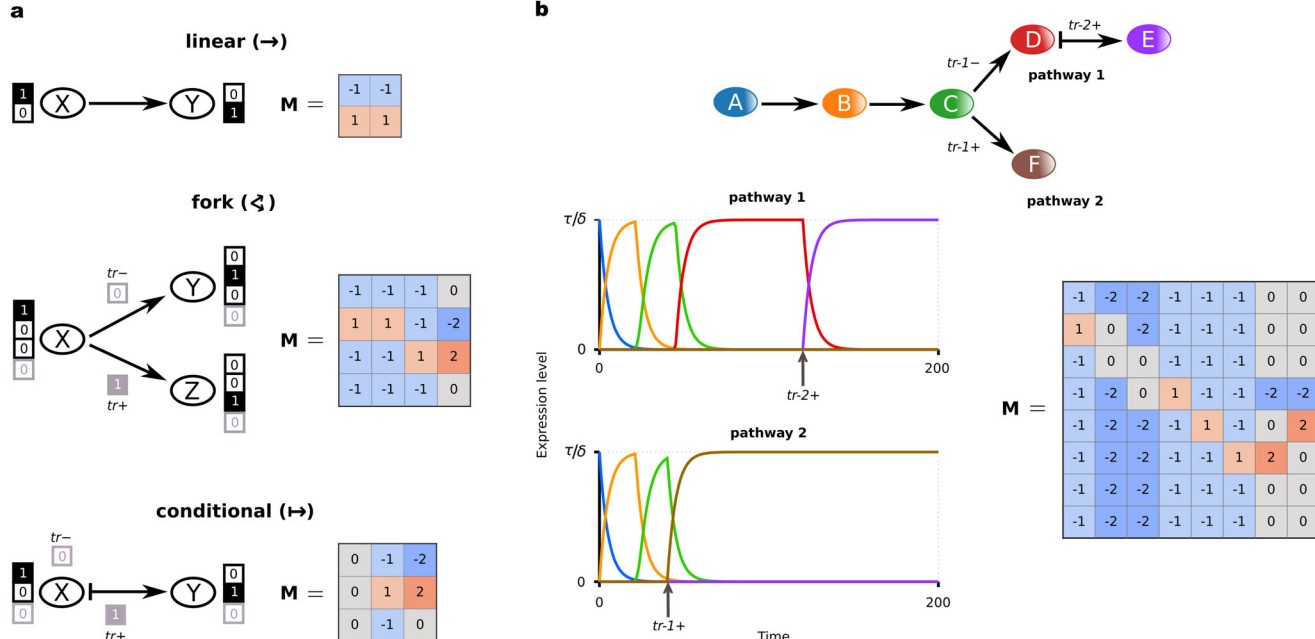

**Fig. 8 | Construction of regulatory program matrices in the AGRN framework.**
**a** Three elementary stage transition types of the model. Linear (upper), fork (middle), and conditional transitions (bottom). Uppercase letters with rounded background represent developmental stages with the corresponding expression profile of the developmental stage vectors in which genes with an on, or off state are indicated by value 1, or 0, respectively. Black outline of the squares denotes stage-specific genes; gray outline refers to triggers (*tr*). The elements of the corresponding regulatory matrix **M** indicate the nature of the pairwise regulatory interactions (negative: repressor, positive: activator, zero: neutral). **b** Illustration of the model functionality on a simple differentiation hierarchy. Due to the fork transition in stage C, there are two possible developmental pathways depending on *tr-1*. C → D is the default pathway that needs no trigger, while C → F is the triggered branch that the

differentiation process follows, if the *tr-1* trigger is on. The conditional transition between D and E stages requires a second signal (*tr-2* +). We used minimal expression representation: stage A corresponds to a stage vector in which the first element is 1, while in stage F the 6th value is 1; the 7th and 8th values of the stage vectors correspond to triggers *tr-1* and *tr-2*, respectively. The lower left panel shows the system's state as a function of time, as measured by the expression levels of the stage-specific genes in case of the two possible developmental pathway realizations (line colors correspond to the colors of the stages as shown in the upper panel). Arrows denote the time of the induction of the trigger signals. The regulatory program matrix **M** corresponding to this system is shown in the rightmost panel. This matrix is derived by a combination of the elementary transition rules depicted in (**a**) and defined in Eqs. 4–6. We used the standard parameter set (see Methods).

corresponding matrix is the dyadic product of the two expression vectors as:

$$\mathbf{M}_{X \dashrightarrow Y} = (2\mathbf{y} - \mathbf{1}) \circ \mathbf{x}, \tag{2}$$

where **1** denotes the all-ones vector. This formula makes intuitive sense; the right-hand term selects the genes implying a regulatory effect (the expressed ones in the present stage), whereas the left-hand term determines the sign of the regulation (depending on the desired high or low expressions in the target stage). In the following we denote this heteroassociative rule by X ⇢ Y. In the special case of X = Y, the X ⇢ X transition implies autoassociativity, rendering the given stage a stable point of the dynamics. This can be described by the following matrix

$$\mathbf{M}_{X \dashrightarrow X} = (2\mathbf{x} - \mathbf{1}) \circ \mathbf{x}. \tag{3}$$

These two associative rules will serve as elements of the functional building blocks of the described GRNs.

Based on these two main associative rules and due to considerations described in the main text, three biologically important transition types should be distinguished; autonomous transition between two stages, fork- and conditional transition. The combination of these transitions allows us to describe almost any kind of biologically plausible interaction topologies. Being components of a network, these transitions are not independent, because each internal stage is involved in at least two transitions (as a departure and a target stage). This poses challenges as any internal stage should be fully expressed, but must not be stable as the system has to go to the next stage. These seemingly contradictory issues can be resolved by proper combinations of auto- and heteroassociative rules as follows.

**Linear transition**
The simplest task is when the gene expression changes from X to Y without any external or internal triggers. Initiating a change from X toward Y requires an X ⇢ Y heteroassociative rule (directionality condition), but this rule itself does not guarantee that the trajectory actually approaches Y. The desired target stage Y must also be autoassociative (attractivity condition) (see ref. 31), that guarantees the high level of expression of the state. The sum of these two conditions yields the regulatory matrix that realizes this transition (Fig. 8a):

$$\mathbf{M}_{X \to Y} = \mathbf{M}_{X \dashrightarrow Y} + \mathbf{M}_{Y \dashrightarrow Y} = \underbrace{(2\mathbf{y} - \mathbf{1}) \circ \tilde{\mathbf{x}}}_{X \dashrightarrow Y} + \underbrace{(2\mathbf{y} - \mathbf{1}) \circ \tilde{\mathbf{y}}}_{Y \dashrightarrow Y}. \tag{4}$$

**Fork transition**
Developmental or differentiation processes are flexible; the gene expression patterns may follow different pathways depending on internal or external conditions. This type of transition can be expressed by fork transitions in the present framework. Depending on the on/off state of a trigger, an X stage may develop into either stage Y or stage Z. This can be considered as a X → Y linear transition by default, which becomes an X → Z transition if the control gene is expressed in stage X (denoted by X' stage). Therefore, on one hand the activation of the control gene must turn off the X ⇢ Y heteroassociativity, and on the other hand it must turn on the X ⇢ Z heteroassociativity. Similar to the considerations presented for the linear transition case, both the Y and Z stages must also be autoassociative. Incorporating these requirements into the regulatory matrix expression

(Fig. 8a) yields:

$$
\begin{aligned}
\mathbf{M}^{\mathbf{s}}_{X \rightleftarrows Y, Z} &= \mathbf{M}_{X \to Y} + \mathbf{M}_{X'-X \to -Y} + \mathbf{M}_{X'-X \to Z} + \mathbf{M}_{Y \to Y} + \mathbf{M}_{Z \to Z} = \\
&= \underbrace{(\mathbf{2y}-\mathbf{1}) \circ \tilde{\mathbf{x}}}_{X \to Y} - \underbrace{(\mathbf{2y}-\mathbf{1}) \circ \mathbf{s}}_{X'-X \to -Y} + \underbrace{(\mathbf{2z}-\mathbf{1}) \circ \mathbf{s}}_{X'-X \to Z} + \underbrace{(\mathbf{2y}-\mathbf{1}) \circ \tilde{\mathbf{y}}}_{Y \to Y} + \underbrace{(\mathbf{2z}-\mathbf{1}) \circ \tilde{\mathbf{z}}}_{Z \to Z} = \\
&= (\mathbf{2y}-\mathbf{1}) \circ \tilde{\mathbf{x}} + 2(\mathbf{z}-\mathbf{y}) \circ \mathbf{s} + (\mathbf{2y}-\mathbf{1}) \circ \tilde{\mathbf{y}} + (\mathbf{2z}-\mathbf{1}) \circ \tilde{\mathbf{z}}
\end{aligned}
\tag{5}
$$

where for sake of notational simplicity we introduce $\mathbf{s} = \mathbf{x}' - \mathbf{x}$ which stands for the expression vector of the trigger (composed of zeros except for the trigger element). Note that the alternative transition is implemented by giving the rules of the alternative pathway relative to the default; $X' - X$ difference leads to $Z - Y$ difference. The autoassociative terms on stages Y and Z ensure the stability of the final stages.

## Conditional transition

Developmental transitions are often triggered by some external or internal cues. Depending on the on/off state of a trigger, an X stage may develop into an Y stage or remain in X. This can be considered as a stable X stage by default, which becomes an $X \to Y$ transition when the trigger is on (X' stage). The expression of the control gene turns off the $X \dashrightarrow X$ auto-associativity, and it turns on the $X \dashrightarrow Y$ heteroassociativity simultaneously. By adding the $Y \dashrightarrow Y$ autoassociative term to warrant the stability of the final stage, we obtain the regulatory matrix (Fig. 8a) expression:

$$
\begin{aligned}
\mathbf{M}^{\mathbf{s}}_{X \nrightarrow Y} &= \mathbf{M}_{X'-X \to -X} + \mathbf{M}_{X'-X \to Y} + \mathbf{M}_{Y \to Y} = \\
&= \underbrace{-(\mathbf{2x}-\mathbf{1}) \circ \mathbf{s}}_{X'-X \to -X} + \underbrace{(\mathbf{2y}-\mathbf{1}) \circ \mathbf{s}}_{X'-X \to Y} + \underbrace{(\mathbf{2y}-\mathbf{1}) \circ \tilde{\mathbf{y}}}_{Y \to Y} = \\
&= 2(\mathbf{y}-\mathbf{x}) \circ \mathbf{s} + (\mathbf{2y}-\mathbf{1}) \circ \tilde{\mathbf{y}}
\end{aligned}
\tag{6}
$$

where $s = \mathbf{x}' - \mathbf{x}$ stands for the expression vector of the trigger as before.

Note that in all types of transitions the target stages are stabilized by an autoassociative term that ensures the high level of expression of the respective stage and the stability of this high level if it is the last state of a series of transitions. This autoassociative step can be placed before the departure stage or can be appended to the target stage; this is a matter of definition (we used the latter), but it is important to avoid duplication. A conditional transition can be considered as a special fork transition, where one branch leads from X to Y, and the other branch leads back to X. Figure 8 illustrates the basic building blocks of the three elementary stage transitions considered in the model and the functionality of the derived regulatory program matrix for a simple artificial differentiation topology. A step-by-step guide for building the simple model system described in Fig. 8 can be found in Supplementary Note 1.

## Expression length optimization

Since the basic parameter set assumes the same $\delta$ degradation rate for all gene products, by default, the expression lengths of different stages are almost the same. Therefore, in the cell-cycle model (Fig. 3), we adjusted the expression lengths to mimic the empirically observed relative stage lengths. For this purpose, we used an evolutionary algorithm (for details, see: Supplementary Note 3) by which we set different decay rates for different gene products, resulting a decay rate vector $\boldsymbol{\delta}_i, (i = 1, \ldots, N)$, where $N$ is the number of gene products. Our analysis suggests that using this approach one can obtain arbitrary phase lengths. Moreover, our simulations indicate that the $\delta$ decay rate can also be used as a scaling parameter for the characteristic time of the transitions (the time difference between two consecutive expression level peaks of two stage-specifically expressed gene products, see Fig. S2).

## Robustness analysis

Under the first perturbation scenario, i.e., in the regulation strength perturbation analysis (i), we investigated the robustness of the functionality of the regulatory program matrices against multiplicative and nullifying perturbations. For these analyses, we used gene expression data from the following three systems: human hematopoiesis (Supplementary Data 1), human cell cycle (Supplementary Data 2), and *C. elegans* P5.p vulval precursor cell (VPC) differentiation (Supplementary Data 4). In case of multiplicative perturbations, we perturbed random 1% of the elements of the respective matrix of the system according to the following: $m'_{ij} = m_{ij} \cdot \mathcal{N}(1, \sigma)$, where $m'_{ij}$ is the perturbed element and $\mathcal{N}(1, \sigma)$ is a random number drawn from a normal distribution with unit mean and $\sigma$ standard deviation. To avoid the biologically implausible change in the sign of the regulations if the random number is less than zero, we use zero instead of minus values of the distribution. In case of nullifying perturbations, a given proportion of the total elements of an $\mathbf{M}$ matrix was set to zero assuming that some mutations destroy particular binding sites, leaving the rest unmodified. The performance of the system was measured by the fraction of successful simulations, i.e. the fraction of the cases, when the system followed a predetermined pathway without errors, and all involved gene states were clearly expressed with at least 0.95 Pearson correlation (computed between the $\mathbf{p}(t)$ expression vector and the stage-specific developmental stage vector). The target was the P5.p VPC→vulA pathway in the *C. elegans* P5.p vulval precursor cell differentiation model (Fig. S4a), and the Meso-derm→CFU-E pathway in the human hematopoiesis model (Fig. 2a). In the human cell cycle model (Fig. 3), simulations were considered to be successful, if the cyclic dynamics was sustained and the system did not enter the apoptotic pathway (Fig. 3c). We made 10000 repeats for each investigated value of the standard deviation $\sigma$ and for each investigated proportion of nullified elements.

Under the second perturbation scenario, i.e., in the misexpression analysis (ii), we analyzed the effects of the mistimed expression of cellular identity determining key genes on the functionality of the hematopoietic system (Fig. 7a, see Fig. 2b right panel). The performance of the system in this case was measured by Pearson correlation coefficients between the $\mathbf{p}(t)$ expression vector (vector for the actual dynamical expression state of the genes) and the developmental stage vectors.

## Publicly available data

The human hematopoiesis dataset (Supplementary Data 1) consists of developmental stage vectors for the cellular stages of the hematopoietic hierarchy. These vectors include binary expression states for 15 key genes whose differential expression is thought to be a major determinant of the cellular identity in differentiating hematopoietic cells[39,68]. For 14 of these genes (*GATA-1, GATA-2, PU.1, SCL, Bra, Flk-1, Runx1, VE-cadherin, c-myb, NF-E2, c-kit, EKLF, EpoR, Fli-1*), stage-specific expression was obtained from ref. 39, and for *BMP4*, from refs. 69,70. Three-membered regulatory subcircuits extracted from the regulatory matrix of this system are shown in Supplementary Table 1.

The human cell cycle (CC) dataset (Supplementary Data 2) is based on a gene expression profiling meta-analysis[71]. In this dataset, we defined binary expression states for the 48 high confidence CC genes that have been identified in at least three of the five primary source CC datasets[44,72–75] which the original meta-analysis[71] considered, and their expression states were determined identically with respect to each stage in all of these datasets (the apoptotic process is simply represented by the expression of an apoptosis-specific gene).

The *C. elegans* embryonic development dataset (Supplementary Data 3) includes gene expression information on 2435 genes (considering only the non-unique ones) corresponding to 1046 differentiation stages. These data were collected from refs. 76,77. after which we fused the two datasets, filtering out genes that were not present in either of them. The considered stages are those from the early development of *C. elegans* embryonic cell lineages, starting from the P0 cell with 454 fork and 137 linear transitions (Fig. 4a).

We also used *C. elegans* as a model to test the performance of the suggested AGRN framework on a system that implements organogenesis (i.e., vulva development from the P5.p and P6.p vulval precursor cells; see Supplementary Data 4 and Supplementary Data 5, respectively). For this

analysis, we collected gene expression data from refs. 78–80, results of the detailed analysis are shown in Supplementary Note 5.

## Reporting summary

Further information on research design is available in the Nature Portfolio Reporting Summary linked to this article.

## Data availability

External data sources for Figs. 2, 3, 4, 6, 7 and Figs. S1, S3, S4 and Supplementary Table 1 were assembled into Supplementary Data 1-5 and are provided with the paper. All further data supporting the results and the conclusions are included within the article and the corresponding publicly available repository[81].

## Code availability

Software for simulation and visualization were written in C++, Bash and R. Scripts, required software packages, and instructions are available at https://zenodo.org/records/10556585[81].

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

## Acknowledgements

Supported by the National Research, Development and Innovation Office through contracts Élvonal KKP129848, OTKA K141064 and the Templeton World Charity Foundation through "Learning in evolution, evolution in learning" award - TWCF0268. A.S. received support from the Hungarian Academy of Sciences through Bolyai János Research Fellowship program. M.P. and D.V. received support from the ELTE Eötvös Loránd University through Hungarian state PhD scholarship. We thank Balázs Könnyű for helpful discussions, as well as Mauro Santos for comments on the manuscript.

## Author contributions

Conceptualization: A.S., P.S. Methodology: A.S., P.S. Investigation: A.S., D.V., M.P., P.S. Visualization: M.P., D.V., A.S., P.S. Funding acquisition: A.S.,

E.S. Project administration: A.S. Supervision: A.S., E.S. Writing—original draft: A.S., P.S., E.S., M.P., D.V. Writing—review and editing: M.P., D.V., G.J., A.S., E.S.

## Funding

## Competing interests
The authors declare no competing interests.
