## [Peer Review File · Communications Biology]

Reviewers' comments:

Reviewer #1 (Remarks to the Author):

Review

Neural network model of cell differentiation on Waddingtonian Landscapes

Paczkó et al. propose a neural network inspired modelling technique of gene regulatory networks that control cell differentiation processes. The AGRN proposed here stores the stage-specific gene expression profiles as memory patterns and

can dynamically regulate the expression states of individual genes.

The authors apply their method to three different biological systems: hematopoietic cell differentiation, human cell cycle dynamics and *C.elegans* embryonic development.

I find the study interesting however, I have several concerns which in my opinion should be addressed before considering this study for publication.

1. I think the AGRN method needs to be better presented. I understand that some aspects of the modelling technique come from previous studies from the same group however I think it is important to be able to grasp what is done in this study without needing to read another two papers. I strongly think that the authors should present a very simple example and take the reader through all the details. Since they already investigated hematopoiesis, I suggest looking into May et al. 2013 paper from Enver group, use the data from there, apply their framework to the GRN in the paper with only three nodes while one could consider only two steady states. Hopefully with this very small example and all the details presented one should be able to understand the new method even if the reader does not have a computational background.

2. Related to 1, I have a few things that are still unclear to me:

2.1 what is the input and what is the output of the modelling technique?

2.2 Looking at the results looks like the Pearson correlations between the p vector and the differentiation data are the output. At the same time it feels like the stage specific data is used for training or? So basically, one trains with the same data that one checks for correlations?

2.3 Moreover the only implementation that I could see of a GRN is in the M matrix, however the authors say: "the network expression here must not be confused with a gene regulatory network represented by an M regulatory program matrix". So is it M the predicted GRN or there is something that I missed here?

2.4 If M is also an output from the model, did the authors compare the obtained GRN with any existing GRN (from other studies) shown to govern the development processes modelled here? If not than the simple example suggested at point 1 should be used for this as well.

3. I find the title a bit too misleading, the modelling technique is Neural Network like but not really an

NN while the Waddington landscape feels like it comes as an afterthought not something that one focuses on.

4. I suggest the authors to put their work in a wider computational context not just their own work. They should look into studies that aimed at implementing new similar methods for similar biological applications e.g. Andersson et al. 2022 – CELLoGeNe ...

5. I am also missing the explanation of the predictive power of AGRN method. I mean it looks like it reproduces the known developmental stages for various biological systems but what does it predict? The main point of a model is its prediction power. Maybe the authors can give some examples on how to use their method to answer important biological questions even if they do not achieve this here.

6. The authors used the following standard parameter set: $\tau = 1$, $\delta = 0.2$, $\omega = 50$, $\xi = 0.05$. Why these values? Are they extracted from some data? If not, could the authors conduct a sensitivity and robustness analysis related to these parameters? At least we get to know how they influence the results.

7. In the figures where the Time is one of the axes, what is the time unit is it mins, hours? Is this simulation time or experiment time? If it is simulation how does this connect to the actual time for the modelled development processes?

8. In figure 4 there are some pathways that are not achievable, however the authors state that they can fix this by adding more forks to the system. What is the right number of forks? Can one overfit?

Reviewer #2 (Remarks to the Author):

The work by Paczkó et al presents an interesting modeling approach based on associative neural networks to describe gene regulatory networks. However, much of the model development is unclear: the structure of the neural network, its correspondence with biology, the process of predicting experimental data ...all need be extensive clarification.

- Lines 47-50: “Given a set of desired stable states (autoassociativity) or stage-pair transitions (heteroassociativity), the regulatory network of a given differentiation topology can be analytically determined by simple algebraic operations in the form of a regulatory weight matrix.”

- The structure(s) of the network need to be more clearly elucidated.
- What’s the biological interpretation of the ANN? Is it supposed to represent the GRN hierarchy?
- Do the nodes / neurons in the network represent individual genes / Transcription Factors?
- Or does each layer represent the same ensemble of genes?
- How many nodes in each layer of the ANN?
- How many layers in the ANN?
- Are there two separate artificial neural networks / ANNs (auto-associative and hetero-associative)? Do they both have the same weight matrix?
- Is a particular gene represented in multiple layers?
- Is the overall GRN given by the collection of the weights in the multiple layers of the network?

- Lines 50 – 53: “the gene expression values across the differentiation stages will be ultimately determined by the regulatory matrix and a shared activation function that nonlinearly maps the summed regulatory effects (weights) of all genes into expression values.

- Why would an activation function be “shared” by multiple genes feeding into a downstream gene?

- “...elements representing the on/off states of different genes...”

- Is the GRN representation Boolean?

- Hematopoiesis Figure (Fig 1): What’s the accuracy of prediction? Are time courses being predicted? Why only one gene per cell state?

- Lines 101-102: “.... we defined stage-specific gene expression profile vectors”

- What does this mean? Aren’t the vectors essentially the data? Is the data bulk RNA-Seq? Why not use single-cell data?

- Lines 103-105: “the differentiation topology we consider here consists of a moderate number of nodes and a combination of a few fork and linear transitions, and one conditional transition, where the nodes illustrate the potential cell differentiation stages...”

- How is the decision about a “moderate number of nodes” made? How is this extracted from the data?
- Lines 114 to 118: Model performance
- Not clear to me how the Pearson correlation coefficients are calculated / shown? Fig 1b seems to show the time courses of stage-specific representative genes. Why would the correlation vary with time?

- From Supp Discussion 2, “the definitive endothelial cell state has the strongest attractor property”
- Is this from simulation? Actual data? Not clear.
- Why should the definitive EC state be “the lowest lying valley in the epigenetic landscape of hematopoiesis”? What about the other end stages?

- The distinction of default / autonomous vs. triggered / conditional trajectories is unclear. Isn't default / autonomous development itself a result of multiple triggers?

The further major corrections that mostly reflect the Reviewers' concerns are:

- The title of the manuscript (Reviewer #1) has been changed to: “*A neural network-based model framework for cell-fate decisions and development*”.
- We have rewritten the abstract to fit the 150-word limit.
- The “Robustness against perturbations” section has been moved from the Supplementary Material to the Results section of the main text, because we felt that this analysis was relevant to characterize the general behavior of the AGRN framework.
- We added a new paragraph in the Introduction to place the AGRN model framework in a more general modelling context (Reviewer #1).
- We extended the Discussion to explore the biological consequences of our results in a more detailed and accessible narrative (both Reviewers).
- As a result of the above extensions, the reference list increased by 20 newly referred articles.

Reviewers' comments:

Reviewer #1 (Remarks to the Author):

Review Neural network model of cell differentiation on Waddingtonian Landscapes

Paczkó et al. propose a neural network inspired modelling technique of gene regulatory networks that control cell differentiation processes. The AGRN proposed here stores the stage-specific gene expression profiles as memory patterns and can dynamically regulate the expression states of individual genes. The authors apply their method to three different biological systems: hematopoietic cell differentiation, human cell cycle dynamics and *C. elegans* embryonic development.

I find the study interesting however, I have several concerns which in my opinion should be addressed before considering this study for publication.

We thank for a careful consideration of our work and try to convince you about the importance of the paper below.

1. I think the AGRN method needs to be better presented. I understand that some aspects of the modelling technique come from previous studies from the same group however I think it is important to be able to grasp what is done in this study without needing to read another two papers. I strongly think that the authors should present a very simple example and take the reader through all the details.

We thank the Reviewer for this comment, which we fully agree with. In the revised version, we have provided a more detailed presentation of the AGRN method. The presentation of the model was so concise because the size limit of Nature allowed only so much (the manuscript was transferred from Nature). We hope that the expanded model description and the new figure will make our model easier to understand. The changes we have made are as follows:

To elaborate on the AGRN method; first of all, we have made a new figure (now, it is Fig. 1 in the main text) that shows the schematic workflow of the AGRN framework. In relation with this figure, we have modified the Introduction: “*A developmental stage vector is extracted from empirical data and it represents the gene expression profile of a given stage (Fig. 1a). Note that while developmental stage vectors are (constant) binary valued (on/off) vectors, the time-dependent gene expression vector is continuous valued.*” (lines 147-151) and “*Thus, a regulatory program matrix, the developmental stage vectors and the differentiation topology from which the matrix is constructed serve as the model input (Fig. 1a, b), while the time series of gene expression levels, characterizing the differentiation stages and determined by the regulatory program matrix and the triggers, are the model output (Fig. 1c, d). For a detailed mathematical description of the model, see Methods.*” (lines 157-162). Moreover, we have moved a figure from the previous Extended Data section to the Methods section (present Fig. 8) which shows the basic building blocks of our modeling techniques and the regulatory program matrix of a simple, instructive network. The detailed derivation of the regulatory matrix \mathbf{M} of this simple network can be found in Supplementary Note 1.

We have also expanded the first section of the Supplementary Information (Supplementary Note 1 - Step-wise regulatory program matrix derivation), in which we show step-by-step the construction of the simple network mentioned above.

We hope that with these modifications the process of building an AGRN model can be understood without the relevant preceding articles (e.g. Vohradsky (2001), Szilágyi *et al.* (2020)).

Since they already investigated hematopoiesis, I suggest looking into May *et al.* 2013 paper from Enver group, use the data from there, apply their framework to the GRN in the paper with only three nodes while one could consider only two steady states. Hopefully with this very small example and all the details presented one should be able to understand the new method even if the reader does not have a computational background.

We have analyzed the hematopoiesis dataset from the May *et al.* 2013 paper in which the authors observed the transcription factor governed gene expression program of multipotent cells (MPCs) into neutrophil (N) or erythroid (E) cells

As a first step we identified the differentially expressed genes for the three stages (MPC, N, E) and created binary gene-expression arrays from the results of the post-hoc analyses (FDR p -value adjustment, 5% significance level). We found 16853 differentially expressed

genes (with triggers and stage-specific genes 16858 in total), which makes this model the largest one we have worked on in terms of number of genes, but the smallest one in terms of number of transitions.

For the fork described by our AGRN formalism, one branch is the default branch (it passes through it if there is no trigger), while the other branch is triggered. In the model described in the May paper, both branches of the fork require their own trigger: the first type of trigger (tr-1) causes the system to evolve to the stage E, while the second type of trigger (tr-2) causes the system to evolve to the stage N. In order to model this case, which is not standard in the context of our formalism, we applied *two* conditional transitions at the *same* stage (MPC). In this case, the system leaves the MPC stage only when triggered: with tr-1 to E, with tr-2 to N. Note that the AGRN model is constructed so that only *one* building block (linear transition, fork transition, conditional transition) can be applied at a given stage and therefore the type of transition expected for the model presented in the May paper (two building blocks at the same stage) has not been extensively investigated by us.

Figure 1 in the present text shows that the system performs as expected in this non-standard case: if no trigger is received, it remains in the MPC stage, and if a trigger is received (at $t = 100$), it evolves to the stage set by the trigger.

A characteristic behavior of the system described in the May paper is that the system is self-locking in the E and N stages. This means that injecting tr-2 in the stage E or tr-1 in the stage N has no effect, the system remains in its original state (N or E, respectively). Our model provides partial support for this finding. If the trigger for the other stage (given at $t = 200$) has maximum strength, the system converges to the other stage (Figure 1, top row), contrary to the May article. If the triggers are weaker (less than 80% of the maximum strength), the observed self-locking property of the system is confirmed by our model: tr-2 cannot drive the system out of stage E, just as tr-1 cannot trigger an N→E transition (Figure 1, bottom row).

Figure 1. Double conditional transition in a hematopoiesis model constructed from data from May *et al.* 2013. Model performance is measured in a standard way by Pearson correlation coefficients between the expression vector and the stage-specific developmental stage vectors. Top row: triggers with maximum strength, bottom row: weaker triggers (80% of their maximum).

2. Related to 1, I have a few things that are still unclear to me:

2.1 what is the input and what is the output of the modelling technique?

The inputs of this modeling technique are: *i*) the topology of the developmental process and *ii*) the stage-specific expression patterns (ON-OFF binary vectors) in the different developmental stages. From these vectors and the topology, the regulatory program matrix \mathbf{M} can be constructed according to simple, modular algebraic rules described in Eqs. (4-6). The output of the system is a time series of expressions of all genes governed by the dynamics of Eq. (1). Note that the dynamics is determined solely by regulatory matrix \mathbf{M} which is, however, responsive to external cues (triggers) at fork and conditional transitions (see the relevant sentences at lines 154-162 and at Fig.1 caption).

2.2 Looking at the results looks like the Pearson correlations between the \mathbf{p} vector and the differentiation data are the output. At the same time it feels like the stage specific data is used for training or? So basically, one trains with the same data that one checks for correlations?

We use the Pearson correlation between the current value of \mathbf{p} and the expression patterns of different developmental stages (the stage-specific expression patterns) to measure the performance of the system. As the system progresses along the desired pathway, the correlation between \mathbf{p} and the current stage-specific vector becomes 1 and then decreases. The model uses the stage-specific vectors as input to the regulatory program matrix \mathbf{M} , and the resulting dynamics can drive the system along any pathway defined by the topology and the external triggers. Therefore, we do not "train" the system with the stage-specific

vectors, but use these vectors to construct the matrix **M**. Regarding the input-output question, see the above answer).

2.3 Moreover the only implementation that I could see of a GRN is in the **M** matrix, however the authors say: “the network expression here must not be confused with a gene regulatory network represented by an **M** regulatory program matrix”. So is it **M** the predicted GRN or there is something that I missed here?

The word “network” can be used in two different contexts in this work: the topology of the differentiation process is a network (differentiation stages are the nodes, developmental transitions are the edges), and to describe the direct and indirect gene-gene interactions that comprises matrix **M**. To reduce the possibility of confusion, we used the terminology “differentiation topology” instead of “differentiation network” throughout the manuscript and used the word “network” only in connection with the regulatory networks represented by the **M** matrices. Thanks for drawing our attention to the inconsistency.

2.4 If **M** is also an output from the model, did the authors compare the obtained GRN with any existing GRN (from other studies) shown to govern the development processes modelled here? If not than the simple example suggested at point 1 should be used for this as well.

A regulatory matrix **M** is an input in our model, as it is composed of the stage-specific vectors and determines the gene expression levels (which are basically the output, see: new Fig. 1 in the main text). Since an **M** matrix in our model is a very mathematical construction and, as such, does not intend to describe exact, gene-specific regulatory interactions, but rather it represents composite regulatory effects (direct and indirect gene, TF, protein and epigenetic elements combined), we held the view that the GRN topology comparison – in the strict sense of the term – is not very relevant for this study.

However, we have in fact made a relevant comparison with existing network subcircuits, a quantitative approach to uncover the basic building blocks of networks. In this analysis, we extracted three-membered subcircuits from the regulatory matrix that drives the hematopoietic dynamics in our model (Supplementary Table 1). We found that the most frequent triplet motif (denoted as I. in the table) that can be identified in this network is known as "single-input module (SIM)" in which a regulator X regulates a group of target genes. The main function of this motif in gene regulatory networks is to allow coordinated expression of a group of genes with shared function (Alon, 2007; doi: 10.1038/nrg2102). The second most frequent motif (denoted as II. in the table) is known as "interacting transcription factors that coregulate a third gene". Similar to our analysis, this is the second most abundant motif in an empirical study of integrated cellular networks, where composite motifs formed by protein-protein interactions (PPIs) and gene-gene interactions were analysed (Lotem et. al, 2003; doi: 10.1073pnas.0306752101).

3. I find the title a bit too misleading, the modelling technique is Neural Network like but not really an NN while the Waddington landscape feels like it comes as an afterthought not something that one focuses on.

Our model is, *sensu stricto*, a NN model based on single-layer recurrent artificial neural networks, and can be treated as a special type of Hopfield network.

Although there is no universally accepted definition of the term “Waddingtonian landscape”, we felt that the attractor dynamics underlying our method was closely analogous to the accepted understanding of it. To reduce the possibility of misunderstanding we have changed the title to “*A neural network-based model framework for cell-fate decisions and development*”. We hope we have made it clear.

4. I suggest the authors to put their work in a wider computational context not just their own work. They should look into studies that aimed at implementing new similar methods for similar biological applications e.g. Andersson et al. 2022 – CELLoGeNe ...

We agree with the Reviewer's point of view that the introduction was too brief due to Nature's length restrictions. To provide a more comprehensive computational context for our framework, we have included a complete paragraph in the introduction (lines 48-82). Furthermore, we have cited Andersson *et al.* (2022) in the same paragraph.

5. I am also missing the explanation of the predictive power of AGRN method. I mean it looks like it reproduces the known developmental stages for various biological systems but what does it predict? The main point of a model is its prediction power. Maybe the authors can give some examples on how to use their method to answer important biological questions even if they do not achieve this here.

Thank you for bringing this to our attention!

The AGRN framework provides a mechanistic and phenomenological explanation for signal-driven cell fate decisions during cell specification and predicts the corresponding time series of gene expression levels. We see the novelty of this approach in that while it models the differentiation process as attractor dynamics, it also incorporates sensitivity to external cues (triggers), thereby providing flexibility in cellular identity output.

In relation with this, we also call attention to the fact that our approach is consistent with the notion of multilineage priming, whereby multipotent cells simultaneously exhibit co-accessibility of several lineage programs and have in place transcriptional circuits capable of responding to multiple extrinsic signals (see, e.g., May & Enver, 2013; doi: 10.1016/j.cub.2013.06.054). In our implementation, associative memory storing and retrieval can thus be regarded as they represent multilineage priming, which can generate diverse lineage-committed cell populations in a robust and still flexible manner.

The other potential strength of our model – as compared to, for example, chemical reaction network models or Boolean network models with a large number of parameters – is that it describes the large-scale dynamics of developmental (or regulatory) dynamics with a very few parameters in a simple model context. That is, despite its simplicity, it contains all

possible developmental pathways and has a modular structure. It can also serve as a tool for synthetic biologists to find alternative routes as a result of mistimed triggers (Fig. 5, Fig. S1, Supplementary Note 2), to analyze the stability of a regulatory network, to construct robust network topologies or to find artificial networks with special dynamical properties.

6. The authors used the following standard parameter set: $\tau = 1$, $\delta = 0.2$, $\omega = 50$, $\xi = 0.05$. Why these values? Are they extracted from some data? If not, could the authors conduct a sensitivity and robustness analysis related to these parameters? At least we get to know how they influence the results.

In the present model $\tau = 1$ is used only to fix the time scale, as we have used arbitrary time units. However, τ (together with δ) provides the possibility to fit the system to experimental data with timing. Note that in our human cell cycle model we used different delta values for different genes to demonstrate the model ability to handle different expression timing (the G1 phase is the longest, the M is the shortest, see Fig. 3, the Expression length optimization subsection at the end of the Methods and Fig. S2, Supplementary Note 3). τ / δ determines the level of expressed genes (with our standard parameters is 5), over a wide range of δ and τ the model remains functional. Note that as δ increases, the characteristic time of expression level change decreases, see also Fig. S2, Supplementary Note 3, ω (the steepness of the activation) can be freely increased (the transfer function becomes more rectangular), its decrease has a limit, below a certain value (according to our extensive numerical experiences this value should be at least 15 and can be arbitrarily high) the model is not functional. Fine-tuning of the parameters is not necessary since the qualitative behavior of the model is not affected by the value of the parameters (a general feature of attractor dynamics) and is also an important property of our model.

7. In the figures where the Time is one of the axes, what is the time unit is it mins, hours? Is this simulation time or experiment time? If it is simulation how does this connect to the actual time for the modelled development processes?

In our models (except the cell cycle) we have used arbitrary time unit, which is set by choosing $\tau = 1$. In the cell cycle model, we demonstrate that the model can be fitted to data with developmental stages of different lengths, see the previous response.

8. In figure 4 there are some pathways that are not achievable, however the authors state that they can fix this by adding more forks to the system. What is the right number of forks? Can one overfit?

The green arrows in the figure show the alternative routes that could be achieved by applying post-fork triggers. This model is built according to Eqs. 1, 4-6. To make the unreachable pathways accessible, the topology has to be modified by introducing formal states, see Figure 2 in the present text.

Figure 2. Possible alternative pathway, where one of the formerly "unreachable" stages can be reached by the inserting an additional developmental stage (P2p).

As this is a very mechanistic approach and it needs the (unrealistic) modification of the original developmental topology, we have removed the corresponding sentence (*“Three pathways are unattainable, as they are default cell fates, but these could have been also achieved by adding additional forks to the system.”*, lines 319-320).

Due to the deterministic nature of our model context overfit is not possible.

We thank the reviewer for his/her questions and suggestions and hope that our changes have made the manuscript more accessible.

Reviewer #2 (Remarks to the Author):

The work by Paczkó et al presents an interesting modeling approach based on associative neural networks to describe gene regulatory networks. However, much of the model development is unclear: the structure of the neural network, its correspondence with biology, the process of predicting experimental data ...all need be extensive clarification.

We fully agree with the Reviewer that the modeling technique is not presented in sufficient detail due to the fact that the manuscript was transferred from Nature, which has strong length restrictions. In the present version, we have substantially improved the manuscript as follows.

To elaborate on the AGRN method; first of all, we have made a new figure (now, it is Fig. 1 in the main text) that shows the schematic workflow of the AGRN framework. In relation with this figure, we have modified the Introduction: “*A developmental stage vector is extracted from empirical data and it represents the gene expression profile of a given stage (Fig. 1a). Note that while developmental stage vectors are (constant) binary valued (on/off) vectors, the time-dependent gene expression vector is continuous valued.*” (lines 147-151) and “*Thus, a regulatory program matrix, the developmental stage vectors and the differentiation topology from which the matrix is constructed serve as the model input (Fig. 1a, b), while the time series of gene expression levels, characterizing the differentiation stages and determined by the regulatory program matrix and the triggers, are the model output (Fig. 1c, d). For a detailed mathematical description of the model, see Methods.*” (lines 157-162). Moreover, we have moved a figure from the previous Extended Data section to the Methods section (present Fig. 8) which shows the basic building blocks of our modeling techniques and the regulatory program matrix of a simple, instructive network. The detailed derivation of the regulatory matrix \mathbf{M} of this simple network can be found in Supplementary Note 1.

We have also expanded the first section of the Supplementary Information (Supplementary Note 1 - Step-wise regulatory program matrix derivation), in which we show step-by-step the construction of the simple network mentioned above.

We hope that with these modifications the process of building an AGRN model can be understood without the relevant preceding articles (e.g. Vohradsky (2001a,b), Szilágyi *et al.* (2020)).

- Lines 47-50: “Given a set of desired stable states (autoassociativity) or stage-pair transitions (heteroassociativity), the regulatory network of a given differentiation topology can be analytically determined by simple algebraic operations in the form of a regulatory weight matrix.”

- The structure(s) of the network need to be more clearly elucidated.

Please see our previous response, especially the new Fig. 1 in the main text and the paragraph at the end of the introduction. In short, single-layer associative neural networks (where the input and output vectors are identical) provide the model context, in which we can formulate regulatory networks such as GRN by exploiting the attractor properties of associative networks. We used the well-known Hebb's law of association to construct the elementary building blocks (linear transition, fork transition, conditional transition) of the model, cf. Eqs. 4-6. The simple algebraic sum of these modules results in the regulatory program matrix \mathbf{M} , which governs the system described in Eq. 1.

- What's the biological interpretation of the ANN?

The attractor properties and memory of auto- and hetero-associative neural networks make them suitable as a model framework for describe GRNs (cf. Vohradsky (2001) cited in the main text). Accordingly, in this model framework, we exploit the dynamical properties of ANNs to describe the dynamics of GRNs. Thus, the AGRN framework provides a mechanistic and phenomenological explanation for signal-driven cell fate decisions during cell specification and predicts the corresponding time series of gene expression levels. We see the novelty of this approach in that while it models the differentiation process as attractor dynamics (clearly, the differentiation stages can be characterized by attractor properties), it also incorporates sensitivity to external cues (triggers), thereby providing flexibility in cellular identity output.

Is it supposed to represent the GRN hierarchy?

Our model is a GRN hierarchy in the sense that gene-gene interactions govern the dynamics, the regulatory program matrix \mathbf{M} represents the direct and indirect gene-gene interactions. However, a direct comparison of our results with traditional GRN hierarchies is not possible because, while traditional models only consider direct gene-gene interactions, our model also considers indirect ones. Furthermore, we emphasize that the neural networks we consider (or any of their building blocks) have no direct, physical counterparts in GRNs, but provide the dynamical framework for modelling.

- Do the nodes / neurons in the network represent individual genes / Transcription Factors?

The expressions of different genes are represented by vector \mathbf{p} . An element of this vector (the expression level of a given gene) is jointly determined by the matrix \mathbf{M} and the transfer function $f(\cdot)$, and this element is considered to be a neuron of our single-layer network.

- Or does each layer represent the same ensemble of genes?

- How many nodes in each layer of the ANN?

The number of nodes is the same as the number of genes plus the number of triggers.

- How many layers in the ANN?

As is common in associative neural networks, the ANN is single layered.

- Are there two separate artificial neural networks / ANNs (auto-associative and hetero-associative)? Do they both have the same weight matrix?

We assemble the regulatory program matrix \mathbf{M} from both auto and heteroassociative parts. The heteroassociative part is responsible for the directionality condition (the system tends toward the target state), the autoassociative part is responsible for the stability condition (the high level of expression in the target state). For example, a linear transition is simply the sum of one auto and one heteroassociative transition, see Eq. (4), Fig. 8a and Supplementary Note 1.

- Is a particular gene represented in multiple layers?

Our network is single layered.

- Is the overall GRN given by the collection of the weights in the multiple layers of the network?

Please see our previous responses.

- Lines 50 – 53: “the gene expression values across the differentiation stages will be ultimately determined by the regulatory matrix and a shared activation function that nonlinearly maps the summed regulatory effects (weights) of all genes into expression values.

- Why would an activation function be “shared” by multiple genes feeding into a downstream gene?

The activation function determines whether a gene is expressed or not as a result of gene-gene interactions (direct and indirect) determined by the regulatory matrix \mathbf{M} . If the total activating effect on a given gene exceeds the inhibitory effects by more than a given value (ξ), expression occurs, otherwise it does not. Choosing $\xi > 0$ guarantees that no expression will occur at low levels of activation, which helps to maintain the large-scale stability of the system. The activation function is therefore responsible for this process and is the same for all genes.

- “...elements representing the on/off states of different genes...”

- Is the GRN representation Boolean?

Our system is Boolean only in the sense that, due to its attractor property, genes have two equilibrium states: off (expression level 0) and on (expression level τ / δ). However, unlike Boolean systems, the expression level transition between the two states is continuous.

- Hematopoiesis Figure (Fig 1): What’s the accuracy of prediction? Are time courses being predicted?

Our model predicts the time evolution of expression patterns. The Pearson correlation shows that the system follows the developmental trajectory defined by the triggers, i.e., the expression pattern in each state matches (or very closely approximates) the expression pattern of the current state. The same can be seen in the PCA plots, where the trajectory

passes through (or approaches) the point that characterizes the given expression state, cf. present Fig. 2.

- Why only one gene per cell state?

In Fig. 1c and Fig. 8 in the main text, any “developmental stage” is represented by only one gene, because these figures describe a didactic model of how the system works. In real systems, states are determined by the expression patterns of many genes. In the model of hematopoietic cell differentiation, 15 genes characterize the system (Supplementary Table 2), whereas in the *C. elegans* development, there are 2435 genes (Supplementary Table 6).

- Lines 101-102: “... we defined stage-specific gene expression profile vectors”
- What does this mean? Aren't the vectors essentially the data? Is the data bulk RNA-Seq? Why not use single-cell data?

The gene expression profile vector is simply the expression pattern of a given state of the system. A state is represented as a binary vector, where the 0 values correspond to the non-expressed state of a given gene, while 1 represents the expressed state (cf. Fig. 1a). In this study, these vectors are constructed on the bases of publicly available data.

- Lines 103-105: “the differentiation topology we consider here consists of a moderate number of nodes and a combination of a few fork and linear transitions, and one conditional transition, where the nodes illustrate the potential cell differentiation stages...”

- How is the decision about a “moderate number of nodes” made? How is this extracted from the data?

Thank you for pointing this out, the wording was inaccurate.

For hematopoietic cell differentiation, we used the classical differentiation topology of this process, following Sviers *et al.*, 2006; doi:10.1016/j.ydbio.2006.02.051. The number of cell types (i.e. differentiation states) and their terminology thus reflects those depicted in Fig. 1 as first scheme (i) in Sviers *et al.*, 2006.

However, we agree that the terminology “node” can be misleading here and elsewhere, where the differentiation cell hierarchy is discussed, as it is generally used in connection with networks, and we simultaneously used it for our regulatory networks as well. In order to remedy this issue and avoid potential confusions, we rephrased this part: “*The differentiation topology we consider here consists of 13 differentiation stages (i.e., cell states), which are modeled by a combination of signal-driven binary cell fate decisions (represented by fork transitions) and autonomous linear transitions, and one conditional (signal-driven linear) transition (Fig. 2a).*” (lines 190-194), and those, where network-related expressions such as “node”, “edge”, “differentiation network”, “graph” were confusingly used in connection with cell types. This implied text changes at the captions of present Figs. 2, 4, 5, 8 (for example, we omitted the word “graph” from the expression “*The graph represents differentiation topology...*” in the caption of the current Fig. 5, and lines 519-523 from the Methods section). Now, the word “network” is only used in connection with the regulatory networks represented by the **M** matrices.

- Lines 114 to 118: Model performance

- Not clear to me how the Pearson correlation coefficients are calculated / shown? Fig 1b seems to show the time courses of stage-specific representative genes. Why would the correlation vary with time?

We measure the performance of the system by Pearson correlation between the (time-dependent) expression vector \mathbf{p} characterizing the time evolution of the system and the (constant) developmental stage vectors characterizing the states (which are derived from the literature).

If the dynamics evolves to a particular developmental stage X , then the correlation of the expression vector \mathbf{p} increases with the (fixed) stage specific vector characterizing the expression pattern of stage X . When the system reaches state X , the correlation becomes 1, and as it moves toward to the next state Y , the correlation decreases. The correlation between \mathbf{p} and *all* stage specific vectors was continuously measured. Thus, the curves taking the value of 1 in succession indicate that the system is sequentially go through the expression patterns of successive stages.

- From Supp Discussion 2, “the definitive endothelial cell state has the strongest attractor property”

- Is this from simulation? Actual data? Not clear.

It is derived from the results of three sets of simulations. Since different types of genes (triggers, unique and non-unique genes) have different effects on the behavior of the system, we analyzed the consequences of perturbations of these gene types separately to gain a detailed insight into their effects on the dynamics.

Since triggers are the main drivers of cell-fate development, we first carried out a thorough analysis of the effect of mistimed triggers at different stages. We allowed the system to approach each developmental stage during its normal time evolution, and then applied all triggers at all different (discrete scale) strengths (each in a separate simulation). We recorded the first fully expressed (with at least 95% correlation) stage that the system reached after the mistimed trigger. Since the first fully expressed stage was usually the definitive EC, we found that this stage had the strongest attractor property in the context of mistimed triggers, see Fig. S1a in Supplementary Note 2.

We then analyzed the effect of perturbations of non-unique and unique genes separately. These types of perturbations are more distant from reality, but they can provide additional insight into the dynamical stability of the system. In the case of perturbation of non-unique genes, we set all triggers and unique genes to zero and initialized the system with all possible combinations of zeros and ones of non-unique genes and recorded the first fully expressed state of the system. In this case, the definitive EC also had the strongest attractor property, see Fig. S1c in Supplementary Note 2.

The effects of unique gene perturbation were studied in a similar way to the previous one, except that one unique gene was switched on in each investigation. In this case the definitive EC was the third most frequent fully expressed stage that the system reached first, see Fig. S1b in Supplementary Note 2.

Supplementary Note 2 has been extended by a few sentences to make the process of the perturbation analysis easier to understand.

- Why should the definitive EC state be “the lowest lying valley in the epigenetic landscape of hematopoiesis”? What about the other end stages?

Thank you for pointing out that our description is not clear enough.

From the point of view of both the symbolic Waddingtonian landscape and the energy landscape of the Hopfield networks, the final states of the system, i.e., the end-differentiated cell states in the differentiation topology (for haematopoiesis: definitive EC, primitive hemangioblast, CMP, GLP, BFU-meg, CFU-E) are "deep and large" valleys in the landscape. The mechanistic explanation for this is that, besides the stabilizing autoassociative component (\mathbf{W} , the attractivity condition) for a given end stage, there is no heteroassociative component (\mathbf{V} , the directionality condition) leading to a next state, cf. Eq. (4). Consistent with the above, and in line with our numerical experiences, the end stages have a large basin of attraction, so they occupy a relatively large fraction of the space of expression vectors.

In our investigations, we estimated the size of the basin of different stages by recording which state is the first to be approached at least 95% correlation after perturbing the expression vector at different stages of development. We found that the definitive EC has the largest basin of attraction. Instead of "the deepest valley", it is more accurate to use "the valley with the largest basin", hence we modified the text accordingly: *“In this context, our investigation on the attractor pool sizes in the hematopoietic cell differentiation hierarchy suggests that the definitive endothelial cell stage can be characterized by the largest basin of the landscape, as this is the stage into which the system-level gene expression pattern (the $\mathbf{p}(t)$ expression vector) converges most frequently in response to different mistimed triggers and perturbed genes (Fig. S1). We emphasize, however, that the latter statement is valid only under the assumption of the presence of these disruptive factors, which result in alternative differentiation trajectories, representing available reprogramming pathways (Supplementary Note 2).”* (lines 443–454).

(However, the measured size of the basin is influenced by the fact that the hematopoietic system we have investigated has a small number of stages. Due to the slow relaxation of the system, some stages are often not yet perfectly expressed (above 95%) immediately after the perturbation, and therefore the first perfectly expressed stage is one of the end stages of the system. If there were more stages before the final stage, there would probably be other perfectly expressed stages before the final one - which would make the apparent size of the basin of end stages smaller).

- The distinction of default / autonomous vs. triggered / conditional trajectories is unclear

Each fork has a "default branch", which is the branch that the system will follow in the absence of a trigger, and a "triggered branch", which the system will follow when the trigger is enabled. A new sentence has been added to the Introduction: "*Each fork transition has a "default branch", which is the branch that the system will follow in the absence of a trigger, and a "triggered branch", which the system will follow when the trigger is enabled.*" (lines 129-131). Based on this, there is a trajectory that does not require a trigger at all (it runs on the default branch at every fork), and there is a trajectory that requires a trigger at every branch. To put this distinction also into a more elaborated biological context, we have added a relevant sentence to the Introduction: "*The existence of such a default output has been suggested, for example, in the case of hematopoietic stem cells (HSCs), which, in the absence of instructive signals, are thought to differentiate into macrophages, an evolutionarily ancient, default blood lineage.*" (lines 131-133).

Isn't default / autonomous development itself a result of multiple triggers?

The system is autonomous, the model contains all possible trajectories, but the trajectory that is implemented is chosen by external influences (triggers) or the lack of them.

We thank the reviewer again for his/her questions and suggestions and hope that our changes have made the manuscript more accessible.

REVIEWERS' COMMENTS:

Reviewer #1 (Remarks to the Author):

Paczkó et al. propose a neural network inspired modelling technique of gene regulatory networks that control cell differentiation processes. As mentioned in my first round of review I find the study interesting.

After reading the reviewed version and the answer to review point by point I finally completely understand the study and I think it became suitable for publication. I want to congratulate the authors on the thorough work conducted and I am happy that even I understand it now.

The only comment that stays is that there is no prediction on something striking that the method brought forward when applied to the three different biological systems.

I let the editor decide if the technique with the presented applications is enough for publishing in their journal.

Reviewer #2 (Remarks to the Author):

Dear authors,

Thank you for addressing my questions and comments. I have no further queries.